# Volcanic cloud detection using Sentinel-3 satellite data by means of neural networks: the Raikoke 2019 eruption test case

Ilaria Petracca[1], Davide De Santis[1], Matteo Picchiani[2,3], Stefano Corradini[4], Lorenzo Guerrieri[4], Fred Prata[5], Luca Merucci[4], Dario Stelitano[4], Fabio Del Frate[1], Giorgia Salvucci[1] and Giovanni Schiavon[1]

[1] Department of Civil Engineering and Computer Science Engineering, Tor Vergata University of Rome, 00133, Italy
[2] GEO-K s.r.l., Rome, Italy
[3] GMATICS s.r.l., Rome, Italy
[4] Istituto Nazionale di Geofisica e Vulcanologia, ONT, 00143 Rome, Italy
[5] AIRES Pty Ltd., Australia

*Correspondence to*: Ilaria Petracca (ilaria.petracca@uniroma2.it)

**Abstract**

The accurate automatic volcanic cloud detection by means of satellite data is a challenging task and of great concern for both scientific community and aviation stakeholder due to the well-known issues generated by strong eruption events in relation to aviation safety and health impacts. In this context, machine learning techniques applied to satellite data acquired from recent spaceborne sensors have shown promising results in the last years.

This work focuses on the application of a neural network based model to Sentinel-3 SLSTR (Sea and Land Surface Temperature Radiometer) daytime products in order to detect volcanic ash plumes generated by the 2019 Raikoke eruption. A classification of meteorological clouds and of other surfaces comprising the scene is also carried out. The neural network has been trained with MODIS (MODerate resolution Imaging Spectroradiometer) daytime imagery collected during the 2010 Eyjafjallajökull eruption. The similar acquisition channels of SLSTR and MODIS sensors and the comparable latitudes of the eruptions allow to extend the approach to SLSTR, thereby overcoming the lack in Sentinel-3 products collected in previous mid-high latitude eruptions. The results show that the neural network model is able to detect volcanic ash with good accuracy if compared with RGB visual inspection and BTD (Brightness Temperature Difference) procedures. Moreover, the comparison between the ash cloud obtained by the neural network (NN) and a plume mask manually generated for the specific SLSTR considered images, shows significant agreement. Thus, the proposed approach allows an automatic image classification during eruption events, and it is also considerably faster than time-consuming manual algorithms. Furthermore, the whole image classification indicates an overall reliability of the algorithm, in particular for recognition and discrimination from volcanic clouds.

## 1 Introduction

From the start of an eruptive event, volcanic emissions are composed of a broad distribution of ash particles, ranging from very fine ash (particle diameters, $d < 30$ µm) increasing in size to tephra (airborne pyroclastic material) with diameters from 2 mm up to 64 mm. Larger fragments are also generated which fall out quickly; these and ash with $d > 30$ µm are not considered in this paper. The gaseous part is made mainly of water vapour ($H_2O$), carbon dioxide ($CO_2$) and sulphur dioxide ($SO_2$) gases (Oppenheimer et al., 2011; Shinohara, 2008), and also a liquid part consisting in sulphate aerosol is present. Depending on the eruptive intensity, the volcanic cloud can reach different altitudes in the atmosphere thus affecting environment (Craig et al., 2016; Delmelle et al., 2002), climate (Bourassa et al., 2012; Haywood & Boucher, 2000; Solomon et al., 2011), human health (Delmelle et al., 2002; Horwell et al., 2013; Horwell & Baxter, 2006; Mather et al., 2003) and aircraft safety (Casadevall, 1994).

The detection procedure consists in identifying the presence of certain species in the atmosphere and discriminating them against other species. Thus, volcanic ash detection is related to the discrimination of the areas (pixels in an image), which are affected by the presence of these particles. First evidences about the possibility to detect volcanic cloud by means of remote sensing data arise in the eighties (A. J. Prata, 1989a; A. J. Prata, 1989b). The method used for the detection of volcanic ash particles relies on the ability to discriminate between volcanic clouds and meteorological ice and liquid water clouds by exploiting the different spectral absorption in the Thermal InfraRed (TIR) spectral range (7–14 µm). In this interval the absorption of ash particles with radius between 0.5 µm and 15 µm at wavelength of 11 µm is larger than the absorption of ash particles at 12 µm. The opposite happens for meteorological clouds, which absorb more significantly at longer TIR wavelengths. Therefore, the Brightness Temperature Difference (BTD), i.e. the difference between the Brightness Temperatures (BTs) at 11 and 12 microns, turns out to be negative ($\Delta T_{11\mu m} - \Delta T_{12\mu m} < 0$ °C) for regions affected by volcanic clouds and positive ($\Delta T_{11\mu m} - \Delta T_{12\mu m} > 0$ °C) for regions affected by meteorological clouds.

The BTD approach is the most used method for the volcanic cloud identification. It is effective and simple to apply, even if it can lead to false alarms in some cases, e.g. over clear surfaces during night, on soils containing large amounts of quartz (such as deserts), on very cold or ice surfaces, in the presence of high water vapour content (F. Prata et al., 2001). As already mentioned, the discrimination between volcanic and meteorological clouds is a challenging task, since the region of the overlap of the two objects shows a mixed behaviour not easily recognizable. In these mixed scenarios, the BTD can be negative not only for volcanic clouds but also for meteorological clouds; thus, some false positive results may occur, as the case of high meteorological clouds. False negative results may arise in the case of high atmospheric water vapour content: the water vapour contribution can hide and cancel out the ash particles effects on the BTD, and then the ashy pixels cannot be revealed. In these cases a correction procedure can be applied (Corradini et al., 2008, 2009; A. J. Prata & Grant, 2001). In addition to the described procedures, other algorithms have been developed (Francis et al., 2012; M. J. Pavolonis, 2010; M. Pavolonis & Sieglaff, 2012; Clarisse & Prata, 2016).

For these reasons, it seems appropriate to use advanced classification schemes to address the task of ash detection, such as
approaches which make use of machine learning techniques, avoiding the need to find for each product the best BTD threshold
for creating the volcanic cloud mask manually, which can be a time-consuming process.
For aerosol and meteorological cloud detection, a neural network (NN) (Atkinson & Tatnall, 1997; Bishop, 1994; Di Noia &
Hasekamp, 2018) based algorithm allows the solution of a classification problem. Starting from inputs containing spectral
radiance values acquired in a specific wavelength band, the model generates a prediction in output by assigning to each pixel
of the original image a predefined class. In previous research, neural networks have already shown significant effectiveness in
terms of atmospheric parameter extraction (Gardner & Dorling, 1998) and specifically for volcanic eruption scenarios (Gray
& Bennartz, 2015; Picchiani et al., 2011, 2014; Piscini et al., 2014). A strong advantage of using a NN based approach for
volcanic cloud detection is that once the model is trained on a statistically representative selection of test cases, new imagery
acquired over new eruptions can be accurately (depending on the training phase) classified in near real time allowing significant
advantages in critical situations and in emergency management.
In this work, we developed a NN based algorithm for volcanic cloud detection using Sentinel-3 SLSTR (Sea and Land Surface
Temperature Radiometer) daytime data with a model trained on MODIS (MODerate resolution Imaging Spectroradiometer)
daytime images. This is possible since the two sensors have similar spectral bands and it represents an advantage as there is
currently limited use of SLSTR products for eruptive events. The use of MODIS as a proxy for SLSTR was already successfully
tested in a previous work investigating the complex challenge of distinguishing ice and meteorological clouds (also containing
ice) using neural networks on SLSTR data (Picchiani et al., 2018). As a test case, the Raikoke 2019 eruption has been
considered in this work.

## 2 Case study: the Raikoke 2019 eruption

The Raikoke volcano is located in the Kuril Island chain, near the Kamchatka Peninsula in Russia (48.3° N, 153.2° E). On
June 21, 2019 at about 18:00 UTC Raikoke started erupting and continued erupting until about 03:00 UTC on 22 June 2019).
During this period, Raikoke released large amount of ash and $SO_2$ into the stratosphere.
Figure 1 shows a time-series of 11 µm brightness temperatures (BTs) determined from the Himawari-8 AHI (Advanced
Himawari Imager) sensor at 10-minute intervals for the first 18 hrs of the eruption. With the purpose of searching for high
(cold) vertically ascending clouds due to an eruption, and not of meteorological origin, discrete eruptions were identified by
comparing AHI BTs near the vent with those some distance upwind from the vent. The Himawari-8 time-series shows a
sequence of eruptions (12 in all) and a sustained period of activity between 22:40 of 21 June and 02:10 of 22 June, when the
majority of ash and gas was emitted. The estimated time of an eruption event was determined by examining animated images
and consequently the times of eruptions shown do not always coincide with the coldest cloud-top. It is estimated from the
AHI data that June 2019 Raikoke eruption produced approximately 0.4–1.8 Tg of ash (Bruckert et al., 2022; Muser et al.,
2020; A. T. Prata et al., 2022) and 1–2 Tg of $SO_2$ (Bruckert et al., 2022; Gorkavyi et al., 2021). The amount of water vapour
emitted is unknown, but would have been considerable, as common in most volcanic eruptions (Glaze et al., 1997; McKee et
al., 2021; Millán et al., 2022; Murcray et al., 1981; Xu et al., 2022). These emissions would have led to copious amounts of
water and ice clouds being produced (McKee et al., 2021; Rose et al., 1995), making the composition of the transported clouds
both complex and changing with time.

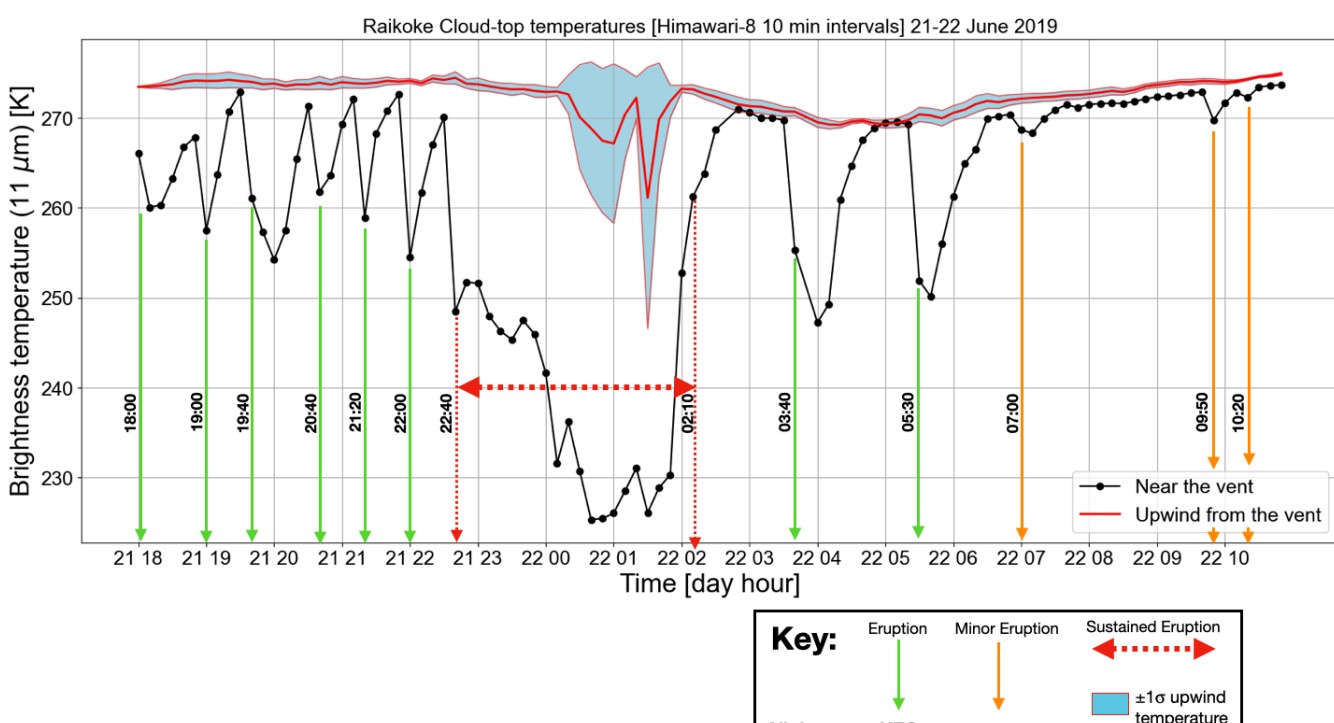

**Figure 1:** Time-series of eruptions from Raikoke during the first 18 hrs of activity. The times of eruptions were estimated from the imagery
and do not always coincide with the coldest cloud tops. (Black line is the average within a box bounded by the latitude/longitude coordinates:
153.25-153.35°E, 48.32-48.42°N.  The red line (upwind) is the average within a box bounded by: 153.10-153.20°E, 48.32-48.42°.

## 3 Instruments

In this section the specifications of the instruments which provide the products used to conduct the research are described. The
MODIS sensor on board Terra and Aqua satellites has been used to set up the training dataset of a NN based model. The
SLSTR sensor on board Sentinel-3A and Sentinel-3B satellites has been used for the application of the aforementioned model.

### 3.1 MODIS instrument

MODIS aboard NASA Terra and Aqua polar orbit satellites is a multispectral instrument, with 36 channels from VIS to TIR
ranging from 0.4 to 14.4 μm, and spatial resolutions of 0.25 km for bands 1-2, 0.5 km for bands 3-7 and 1 km for bands 8-36.

The two spacecrafts fly at 705 km of altitude in a sun-synchronous orbit, with a revisit cycle of about one or two days. Terra
spacecraft was launched in 1999 and its equatorial crossing time is 10:30 am (descending node), while Aqua was launched in
2002 and its equatorial crossing time is 1:30 pm (ascending node).
In our work we used several Terra-Aqua/MODIS products: Level-1A Geolocation Fields (MOD/MYD03) (see (Nishihama et
al., 1997) for details), Level-1B Calibrated Radiances (MOD/MYD021KM) (see (Toller et al., 2017) for details), which has
been used to generate the Brightness Temperatures (BTs), Level-2 Surface Reflectance (MOD/MYD09) (see (Vermote &
Vermeulen, 1999) for details), Level-2 Cloud Product (MOD/MYD06_L2) (see (Menzel et al., 2015) for details).
**3.2 SLSTR instrument**
The Sea and Land Surface Temperature Radiometer (SLSTR) is one of the instruments on board the Sentinel-3A (S3A) and
Sentinel-3B (S3B) polar satellites launched in 2016 and 2018, respectively.
Sentinel-3 is designed for a sun-synchronous orbit at 814.5 km of altitude with a local equatorial crossing time of 10:00 am.
The revisit time is 0.9 days at equator for two operational spacecrafts configuration. The orbits of the two satellites are equal
but S3B flies +/- 140° out of phase with S3A. The basic SLSTR technique is inherited from the technique used by the series
of conical scanning radiometers starting with the ATSR. The instrument includes the set of channels used by ATSR-2 and
AATSR (0.555 – 0.865 µm for VIS channels, 1.61 µm for SWIR channel, 3.74 – 12 µm for MWIR/TIR channels), ensuring
continuity of data, together with two new channels at wavelengths of 1.375 and 2.25 µm in support of cloud clearing for surface
temperature retrieval. The SLSTR radiometer measures a nadir and an along track scan, each of which also intersects the
calibration black bodies and the visible calibration unit once per cycle (two successive scans). Each scan measures two along
track pixels of 1 km (four or eight pixels at 0.5 km resolution for visible/NIR channels and SWIR channels, respectively)
simultaneously. This configuration increases the swath width in both views, as well as providing 0.5 km resolution in the solar
channels.
Our procedure makes use of the SLSTR Level-1 TOA (Top Of Atmosphere) Radiances and Brightness Temperature product
from both platform S3A and S3B, see (Cox et al., 2021) for details of SLSTR Level-1 product.

**4 Methodology**
In this section the adopted methodology is described. The procedure has been developed in MatLab environment and the
source codes are available upon request, as explained in Code Availability section. In particular, the MatLab Deep Learning
Toolbox has been used to implement the NN.
A multilayer perceptron neural network (MLP NN) was trained with MODIS daytime data and then it was applied to Sentinel-
3/SLSTR daytime products, in order to discriminate ashy pixels from others, following the scheme reported in Figure 2.
The MLP NN model (Atkinson & Tatnall, 1997; Gardner & Dorling, 1998) consists in a multi-layer architecture with three
types of layers. The first type of layer is the input layer, where the nodes represents the elements of a feature vector. The second
type of layer is the hidden layer, and consists of only processing units. The third type of layer is the output layer and it represents
the output data, which are the classes to be distinguished and are set to one (that of the chosen class) or zero (all other nodes)
in image classification problems. All nodes (i.e. neurons) are interconnected and a weight is associated to each connection.
Each node in each layer passes the signal to the nodes in the next layer in a feed-forward way, and in this passage the signal is
modified by the weight. The receiving node sums the signals from all the nodes in the previous layer and elaborates them
through an activation function before passing them to the next layer.
The output of the proposed model is the SLSTR image fully classified in eight different species: ash over sea, ash over cloud,
ash over land, sea, land and ice surfaces, liquid water clouds and ice clouds. This approach has been used because of the readily
available time series of MODIS data, the quality of MODIS products (Picchiani et al., 2011, 2014; Piscini et al., 2014) and the
spatial/spectral similarities between MODIS and SLSTR. The SLSTR and MODIS channels which are used in our research
are shown in Table 1, along with the spectral characteristics of the two sensors.
The first step of our procedure consists in generating the training patterns, that is the "ground truth" to be passed to the NN
model during the training phase. This step represents a crucial aspect in building a NN model since the more the training
dataset is accurate and representative of the problem we want to address the more the NN would be efficient in solving that
problem. For this scope, MODIS products have been used as inputs to a semi-automatic procedure for identifying the different
classes to be discriminated by the NN model in the output image. Some of these classes don't exist as MODIS standard
products, for example the ash classes and the ice surface class; for this reason we derived them by means of different operations
in our semi-automatic procedure developed in MatLab. Other classes are instead already present as MODIS standard product,
for example the land/sea mask.

**Table 1:** Correspondence between MODIS and SLSTR channels.

| SLSTR Channel | λ Centre (μm) | Bandwidth (nm) | MODIS Channel | λ Centre (μm) | Bandwidth (μm) |
|---|---|---|---|---|---|
| S1 | 0.554 | 19.26 | 4 | 0.555 | 0.545-0.565 |
| S2 | 0.659 | 19.25 | 1 | 0.659 | 0.620-0.670 |
| S3 | 0.868 | 20.60 | 2 | 0.865 | 0.841-0.876 |
| S4 | 1.375 | 20.80 | 26 | 1.375 | 1.360-1.390 |
| S5 | 1.61 | 60.68 | 6 | 1.64 | 1.628-1.652 |
| S6 | 2.25 | 50.15 | 7 | 2.13 | 2.105-2.155 |
| S7 | 3.74 | 398.00 | 20 | 3.75 | 3.660-3.840 |
| S8 | 10.85 | 776.00 | 31 | 11.03 | 10.780-11.280 |
| S9 | 12.02 | 905.00 | 32 | 12.02 | 11.770-12.270 |



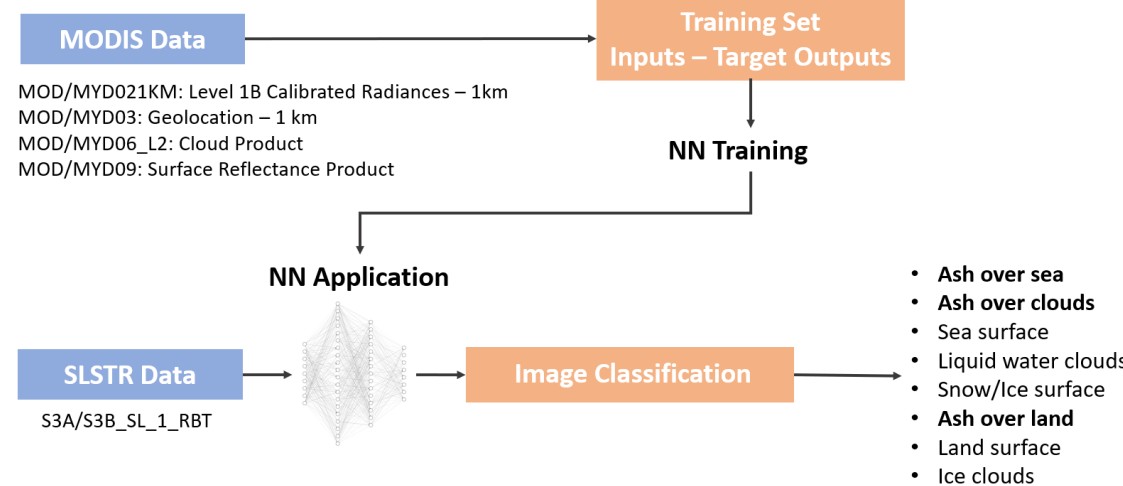


**Figure 2:** Overall diagram of the procedure followed for the classification process with NN model.
The training set from which we extracted the training patterns (i.e. identifying classification classes) consists of nine MODIS
granules acquired over the Eyjafjallajokull volcano area during the 2010 eruption (from May 6[th] to May 13[th]), for a total of
about 5400 patterns for each class available for the training of the model. The single training pattern (i.e.: training example)
corresponds to a single pixel of a specific target class as identified in MODIS images through the semi-automatic procedure
aforementioned, this means that one class is represented by several patterns. In particular, not all the pixels of the considered
MODIS images are contained in the training dataset (i.e.: the ensemble of the training patterns), but only a part of them are
randomly included. The total number of patterns we collected has been divided into three subsets: 75% training set, 20%
validation set, 5% test set. A NN with two hidden layers of was trained and then it was applied to Sentinel-3 SLSTR RBT
(Radiance and Brightness Temperature) Level 1 images collected during the Raikoke 2019 eruption. Table 2 shows the details
of MODIS and SLSTR data used for this work.

**Table 2:** Training set (MODIS) from the Eyjafjallajökull 2010 eruption; Sentinel-3 Raikoke 2019 classified products.

| Date | Time UTC | Platform | Training/Application |
|------|----------|----------|----------------------|
| 6 May 2010  (JD 126) | 11:55 | Terra | Training |
| 9 May 2010  (JD 129) | 12:25 | Terra | Training |
| 11 May 2010 (JD 131) | 12:10 | Terra | Training |
| 11 May 2010 (JD 131) | 12:15 | Terra | Training |
| 11 May 2010 (JD 131) | 13:50 | Terra | Training |
| 11 May 2010 (JD 131) | 14:05 | Aqua | Training |
| 12 May 2010 (JD 132) | 12:55 | Terra | Training |

| | | | |
|---|---|---|---|
| 13 May 2010 (JD 133) | 12:00 | Terra | Training |
| 13 May 2010 (JD 133) | 13:40 | Terra | Training |
| 22 June 2019 (JD 173) | 00:07 | Sentinel-3A | Application |
| 22 June 2019 (JD 173) | 23:01 | Sentinel-3B | Application |

In order to build the NN training patterns a semi-automatic procedure, that exploits MODIS radiances and standard products, has been developed. The MODIS products considered for the extraction of the training patterns are the following:

- MOD/MYD021KM, Level 1B Calibrated Radiances – 1 km, which gives the radiance values for each MODIS band;
- MOD/MYD03, Geolocation – 1 km, used for creating the Land/Sea Mask;
- MOD/MYD06_L2, Cloud Product, containing cloud parameters, used for creating the Cloud Mask;
- MOD/MYD09, Surface Reflectance Product, containing an estimate of the surface spectral reflectance measured at ground level; it is used for generating the Ice Mask;

where "MOD" and "MYD" stands for MODIS-Terra and MODIS-Aqua products respectively.

The semi-automatic procedure for the extraction of training patterns starting from MODIS data basically consists in using MODIS products to create binary "masks" identifying the different species, and then replaces them by "classes". For each element of the class the radiance values ($W/(m^2\ sr\ \mu m)$) are extracted from the MODIS product MOD/MYD021KM. In this way each object is radiometrically characterized. The identification of the ashy pixel is pursued by creating a mask according to specific BTD thresholds (from 0.0 to -0.4 °C) for each MODIS image. For this purpose, the MOD/MYD021KM product has been used to derive the brightness temperatures required to compute the BTD. The MODIS products used for training the model were acquired in near-nadir view only.

The other species are identified using both MODIS Level 1 radiances and MODIS standard products. Once each object/surface has been defined, they are associated with the corresponding class. Then a set of input-output samples for the training phase is generated, where the input consists of the set of radiances measured for the given pixel and the output is a binary vector with value 1 associated with the corresponding class and value 0 for the other classes.

Table 3 shows the classification map legend for each classified product presented in this work, in which eight classes are discriminated, each one representing a surface/object.

**Table 3:** Classification map legend.

| Class ID | Surface/Object | Name | Colour |
|---|---|---|---|
| 1 | Ash over sea | *Ash_sea* | |
| 2 | Ash over clouds | *Ash_cloud* | |
| 3 | Sea surface | *Sea* | |
| 4 | Liquid water clouds | *Cloud* | |

| | | | |
|---|---|---|---|
| 5 | Snow/Ice surface | *Ice* | |
| 6 | Ash over land | *Ash_land* | 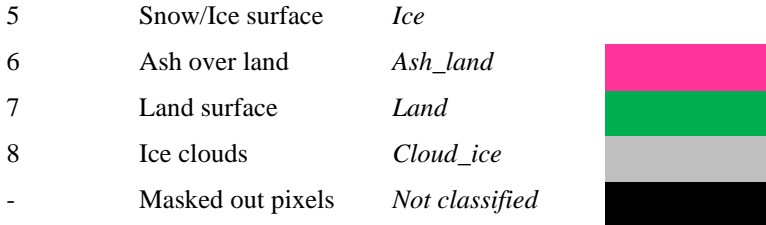 |
| 7 | Land surface | *Land* | |
| 8 | Ice clouds | *Cloud_ice* | |
| - | Masked out pixels | *Not classified* | |

204

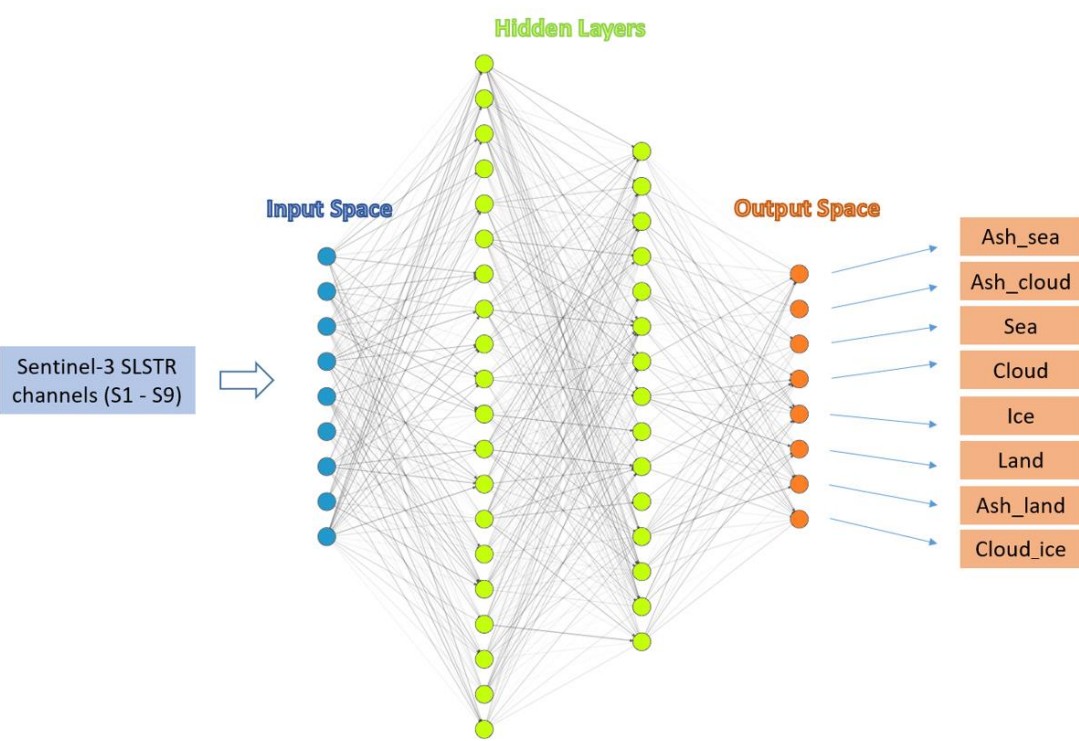

205

**Figure 3:** NN topology for ash detection.

The NN final model consists of nine inputs, which are the radiances in the SLSTR selected channels, while the output space is composed of eight classes, which are the objects/surfaces which the net has to classify. After doing several tests the optimum topology of the NN turns out to be the combination of two hidden layers with 20 and 15 neurons, respectively. For each neuron we set the hyperbolic tangent activation function (Vogl et al., 1988). The final neural network architecture used for ash detection in this work is shown in Figure 3. The proposed algorithm includes a post processing operation in order to avoid false positive results for land and sea classes. This *a-posteriori* filter is applied both to the resulting NN land and sea classes. It allows masking out the pixels which the NN classifies as land/sea which do not belong to the Sentinel-3/SLSTR land/sea mask standard product, which is always available and thus it can be used to increase the precision of the algorithm. The filtered out pixels have been inserted in a class named "not classified", as reported in Table 3.

For classification problems approached with machine learning algorithms, one of the most used accuracy metrics for the performance evaluation is the confusion matrix (Fawcett, 2006), where each predicted output class is compared to the corresponding ground truth considered in the validation dataset. An overall accuracy of 90.9% was obtained at the end of the NN training phase for the proposed neural network model (see Figure 4).

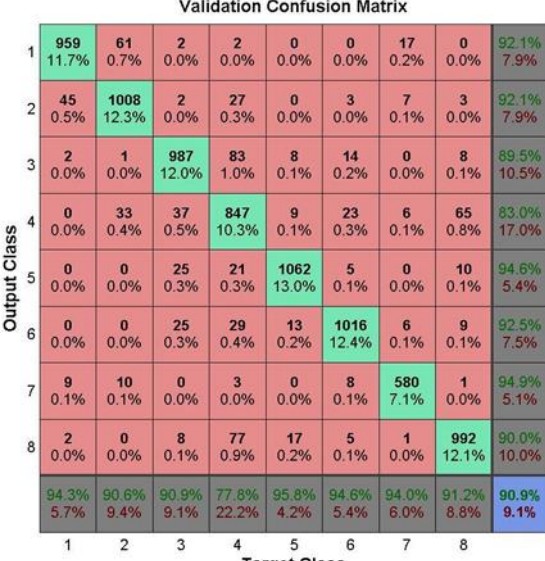

**Figure 4:** Confusion matrix on validation set.

The target class represents the "ground truth" of each class, while the output class refers to the prediction of the NN. The diagonal shows that most of the total of the pixels have been correctly classified (green boxes). The number of pixels incorrectly classified are placed out of the diagonal. False positives (false detection) and false negatives (missed detection) are reported in the last grey column and row, respectively.

The code of the procedure ran with a CPU i7-9850H (6 core, processor base frequency at 2.60 GHz): it takes less than 30 minutes to train the adopted model and few seconds to apply it.

## 5 Results and Discussion

The neural network algorithm previously described was applied to Sentinel-3/SLSTR daytime images acquired on Raikoke during the 2019 eruption. The Sentinel-3A/SLSTR and Sentinel-3B/SLSTR products collected on 22 June 2019 at 00:07 and 23:01 UTC have been considered (see Table 2).

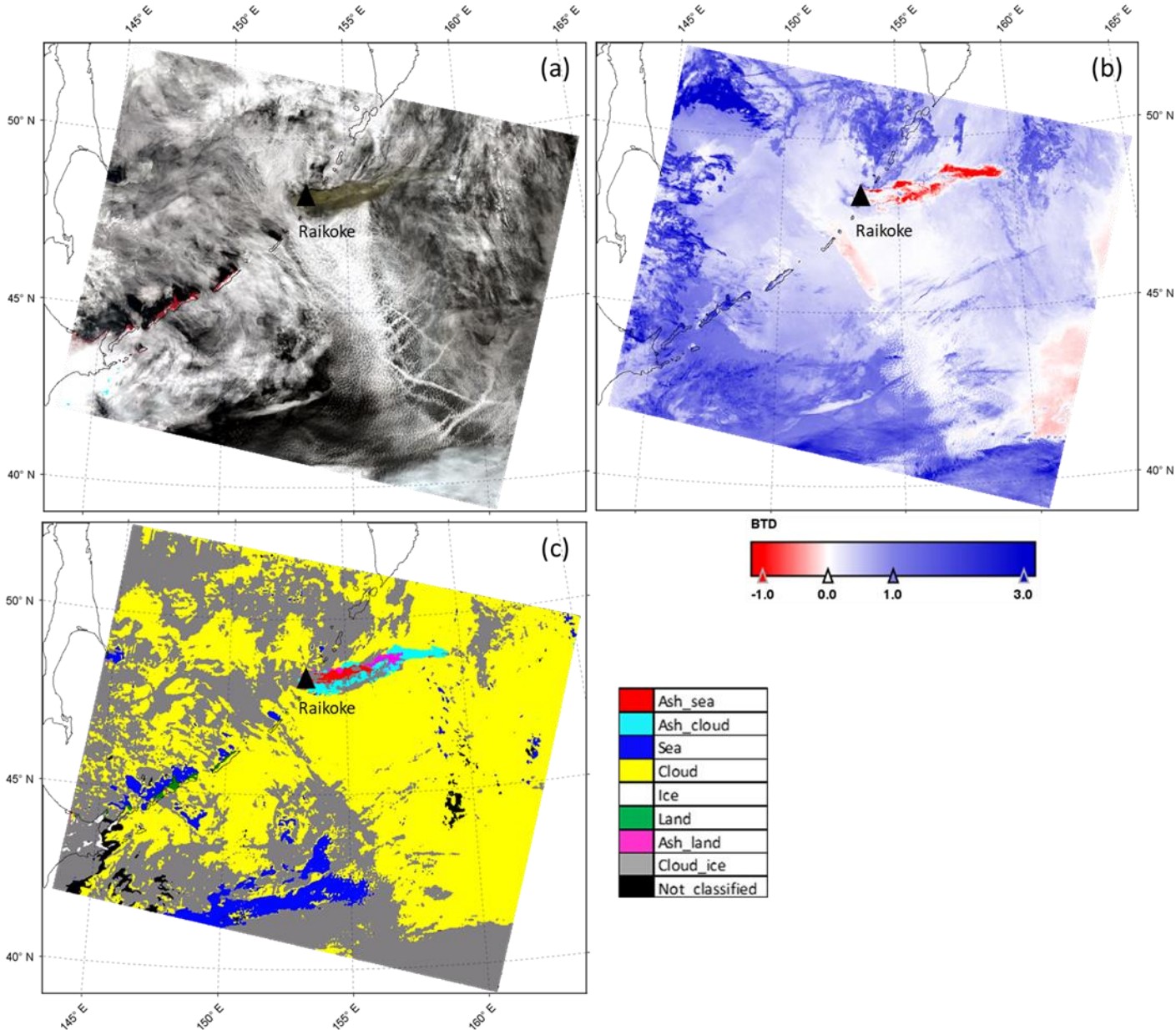

233

**Figure 5:** Sentinel-3A/SLSTR image collected on Raikoke for 22 Jun 2019 at 00:07 UTC, nadir view. (a) RGB; (b): BTD; (c): NN classification.

Figure 5(a) shows the RGB colour composite of the S3A/SLSTR image acquired on Raikoke for 22 June 2019 at 00:07 UTC. The RGB composite has been carried out by considering the SLSTR visible (VIS) channels S3 (868 nm), S2 (659 nm) and S1 (554 nm) for R, G and B, respectively. In Figure 5(b) the BTD map is displayed, where red and blue pixels represent negative and positive BTD, respectively. The BTD is computed by making the difference between the brightness temperature of the

SLSTR thermal infrared channels S8 and S9 centred at 10.8 and 12 μm. The output of the NN classification is shown in Figure
5(c) with the corresponding colour legend, where each colour represents the classified surface/object.

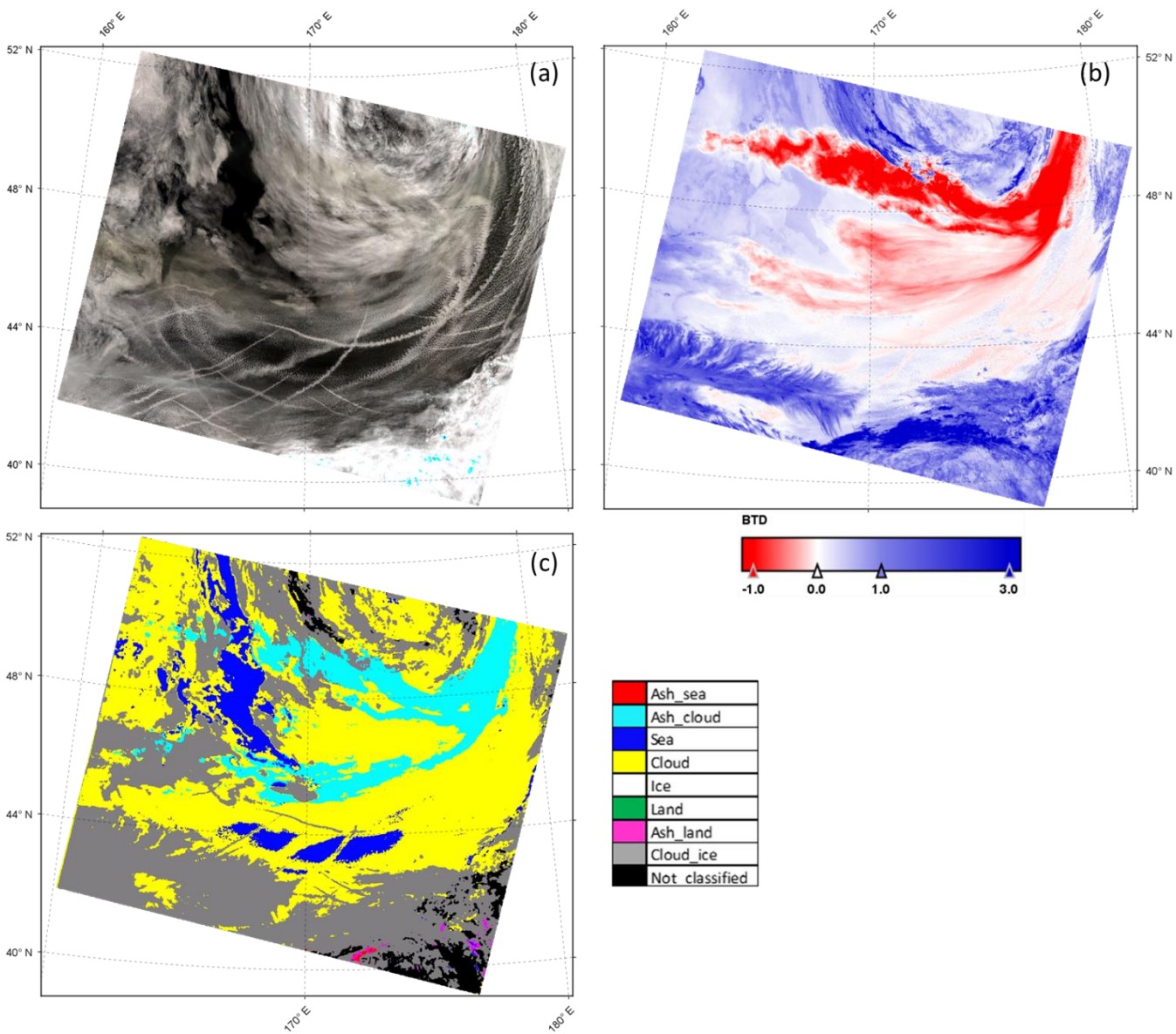


**Figure 6:** Sentinel-3B/SLSTR image collected on Raikoke for 22 June 2019 at 23:01 UTC, nadir view. (a): RGB; (b): BTD; (c): NN
classification.
As Figure 5(a) shows, the RGB composite shows the presence of a wide distribution of meteorological clouds and a significant
signal derived from the volcanic cloud (brown pixels). The BTD (Figure 5(b)), obtained with a threshold of 0 °C, shows the
presence of the volcanic cloud together with a significant number of false negatives (volcanic cloud pixels not identified near
the vents) and false positives (pixels identified as volcanic cloud while actually they are not, see light red pixels below the
volcanic cloud and along the right edge of the scene).
Despite the challenging scenario, the NN algorithm shows its ability to detect the volcanic cloud and to classify the whole
image, by detecting with good accuracy meteorological clouds composed of water droplets (yellow) and ice (grey), sea (blue)
and land (green) surfaces, and volcanic ash clouds, as reported in Figure 5(c). Looking at the cloud masks generated with the
NN algorithm (yellow and grey) and by comparing them with the RGB natural colour composite of the SLSTR product, a high
degree of agreement in terms of spatial features can been observed. From the comparison between NN output classes and RGB
composite we can observe that also land (green) and sea (blue) pixels are properly detected in the areas where they actually
lie.
From a qualitative comparison between the NN plume mask and the RGB composite, we can state that the NN correctly
identifies the volcanic cloud class in the area where it seems actually present, even if some pixels are misclassified as ash over
land (magenta pixels), instead of ash above meteorological cloud. As Figure 5 shows, the NN algorithm is able to detect a
wide volcanic cloud area and more ash, especially in the opaque regions, compared to the BTD approach. In particular the
difference found near the vents can be due to the complete opacity of the cloud. Here the ash cloud optical thickness is so high
that there is no spectral difference and the BTD approach has no sensitivity.
Following the same visualization scheme of Figure 5, the results derived from the application of the trained NN model to the
S3B/SLSTR image acquired on 22 June 2019 at 23:01 UTC are reported in Figure 6. In this second image, all the ashy pixels
are classified by the NN model as ash above meteorological clouds (cyan pixels). This seems reasonable being the scenario
mostly dominated by meteorological clouds, as we can also observe looking at the NN classification, which assigns the
majority of the pixels to the liquid water cloud class (yellow) and to the ice cloud class (grey). The NN classification shows
also the presence of sea pixels (blue), which are located in the same area identifiable using the RGB composite. In this case,
from the RGB composite (Figure 6(a)), unlike what is seen in the 00:07 UTC image, it is not straightforward to identify the
volcanic plume by visual inspection. Indeed, this image was collected about 24 hours later than the previous one and thus the
plume has been transported through the atmosphere and dispersed. A qualitative comparison between the NN classification
(Figure 6(c)) and the BTD map (Figure 6(b)) shows considerable differences between the two methods. The BTD, obtained
with a threshold of 0 °C, identifies a wider area (red pixels) affected by the volcanic cloud with respect to the NN ash mask
(cyan pixels). We can notice that the BTD map includes some aircraft condensation trails (recognizable by the shape in the
RGB composite) in the ash mask, which can be identified as false ash detections. The reasons for these misclassifications are
not fully understood, but may be due to multilayer cloud effects, pixel heterogeneity or viewing angle.
Our results suggest that the NN technique is robust and has shown that it is possible to transfer the NN model from one single
eruption event to others occurring at similar latitudes. However, the complexity of the application suggests that the
generalization of the methodology to all types of eruptions is not straightforward. For example, the change of latitude has an
impact on the characteristics of the atmosphere. At the same time different volcanoes emit different types of ash affecting the
variability of the radiance values detected by the sensors. A possible solution to give to the proposed technique a broader
applicability could be training different NN models for specific latitude belts which can be defined to cover the whole globe.
Overall, we can summarize the main uncertainties and the limitations of the presented model in the following points:

1. model transferability is significantly related to the spatial-temporal data availability for the generation of a training dataset which is statistically representative of all the possible scenarios;
2. lack of standard ground truth data for training and validation phases requires the BTD threshold selection by an operator which prevents the method from being fully objective.

## 5.1 Vicarious validation

The capability of the NN to correctly detect pixels containing ash was validated by making a pixel per pixel comparison with
a reference plume mask generated manually (hereafter MPM) in order to obtain the best accurate *ground truth* as possible in
each SLSTR product. The choice of taking the MPM as reference derives from the lack of ash standard products.
For the image collected at 00:07 UTC the MPM creation was performed by selecting a region around the volcanic cloud
(clearly recognizable as it is at the beginning of the eruption) and then considering only the pixels with 11 µm brightness
temperature < 270 K (see Figure 1). In this case the BTD alone it is not very useful as the high value of the ash optical thickness
of the cloud (especially close to the vent) produces many pixels with BTD values near zero, not distinguishable from adjacent
pixels characterized by meteorological clouds. For the image collected at 23:01 UTC, the identification of the volcanic cloud
is much more difficult due to its larger spread and dilution; in this case the MPM was obtained considering the pixels with
BTD < -0.25 °C, even if probably this choice implies that some ashy pixels were discarded. On the other hand, using an higher
BTD threshold will produce a lot of false positive pixels. In general, the creation of an accurate manual plume mask is time
consuming and case-sensitive and often requires the presence of an operator; so the generation of a volcanic cloud mask with
a fast, automatic and case-independent procedure would be a rather significant improvement.
Because the MPM doesn't distinguish between the different surfaces under the ash cloud, the validation is performed by
considering the total of the ashy pixels detected from the NN (i.e. the sum between *ash_land*, *ash_sea* and *ash_cloud*).
Figure 7 shows the MPM, created as described above, and the comparison between NN plume mask (hereafter NNPM) and
MPM for the S3A/SLSTR image collected on Raikoke for 22 June 2019 at 00:07 UTC (Figure 7(a) and Figure 7(b)) and
S3B/SLSTR image collected on Raikoke for 22 June 2019 at 23:01 UTC (Figure 7(c) and Figure 7(d)).

**Table 4:** NN and BTD volcanic cloud detection accuracies using classification metrics derived from the comparison between the plume
mask obtained from the two approaches and the manual plume mask (MPM) for each SLSTR considered product, respectively.

| Classified Product | Plume mask source | Precision | Recall | F-measure | Accuracy |
|---|---|---|---|---|---|
| S3A/SLSTR at 00:07 UTC | NN classification | 0.709 | 0.683 | 0.696 | 0.993 |
| S3A/SLSTR at 00:07 UTC | BTD < 0 °C | 0.164 | 0.647 | 0.261 | 0.955 |
| S3B/SLSTR at 23:01 UTC | NN classification | 0.773 | 0.657 | 0.710 | 0.935 |

| S3B/SLSTR at 23:01 UTC | BTD < 0 °C | 0.417 | 0.998 | 0.588 | 0.829 |
| --- | --- | --- | --- | --- | --- |


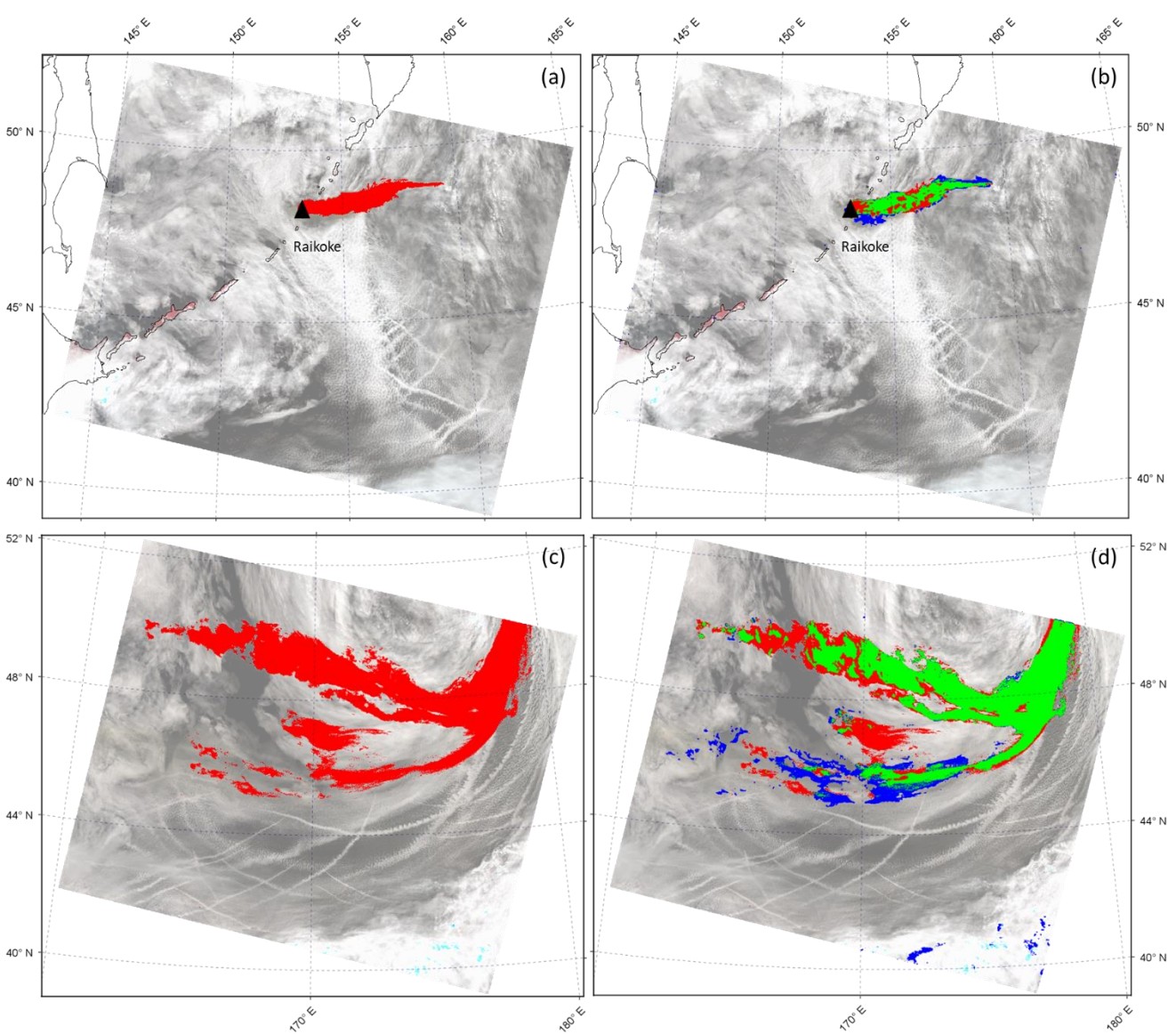


**Figure 7:** Sentinel-3A/SLSTR image collected on Raikoke for 22 June 2019 at 00:07, nadir view (a),(b); Sentinel-3B/SLSTR image collected on Raikoke for 22 June 2019 at 23:01, nadir view (c),(d). (a),(c): red pixels display the manual plume mask (MPM) obtained from the analysis on the specific image; (b),(d): comparison between volcanic ash detected by NN and MPM; green pixels indicate the areas for which both NN and MPM detect ashy pixels, red pixels indicate the areas for which only MPM detects ashy pixels, blue pixels indicate the areas for which only NN detects ashy pixels.




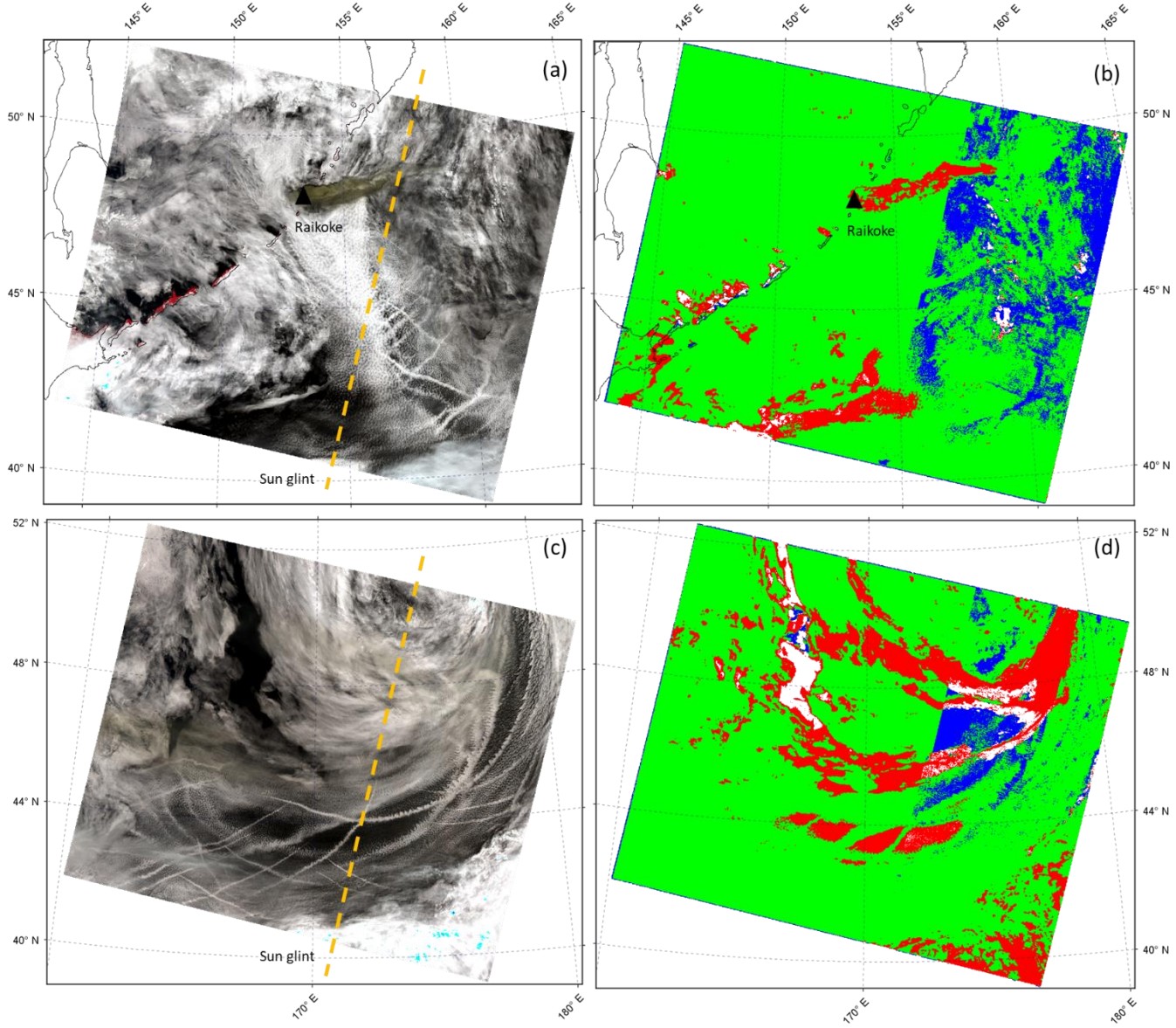


**Figure 8:** Sentinel-3A/SLSTR image collected on Raikoke for 22 June 2019 at 00:07, nadir view (a),(b); Sentinel-3B/SLSTR image collected on Raikoke for 22 June 2019 at 23:01, nadir view (c),(d). (a),(c): RGB view; (b),(d): comparison between cloud mask retrieved by NN and standard Sentinel-3 confidence in summary cloud mask (CSCM); green pixels indicate the areas for which both NN and CSCM detect cloudy pixels, red pixels indicate the areas for which only CSCM detects cloudy pixels, blue pixels indicate the areas for which only NN detects cloudy pixels, white pixels indicate the areas for which both NN and CSCM don't detect cloudy pixels.

326

In relation to the images which display the comparison between NN output and MPM (Figure 7(b) and Figure 7(d)), green areas indicate the pixels for which both the MPM and NN ash masks detect the presence of volcanic cloud, red pixels represent the areas classified as ash only by the MPM; blue pixels are classified as ash only according to the NN model. We can observe

that most of the volcanic cloud is displayed in green for both products (00:07 UTC and 23:01 UTC), indicating good agreement between the two approaches. This is also confirmed by the scores in Table 4, which allow quantitative conclusions on the accuracy of the proposed NN model approach compared to the MPM considered as *ground truth*. The classification metrics considered are precision, recall, F-measure and accuracy (Fawcett, 2006) which range from 0 to 1 (perfect classifier).

The score differences for the two classified products are mainly related to the significant higher number of correctly classified ashy pixels contained in the 23:01 UTC (136435 pixels) with respect to 00:07 UTC (13545 pixels), if compared to the total number of classified pixels in the images which is similar (1614405 pixels for the S3A/SLSTR at 00:07 UTC image and 1701319 for the S3B/SLSTR at 23:01 UTC image respectively). However, the metrics are aligned for both classified data with encouraging values for each index suggesting the reasonability of the results. In particular, the F-measure results of around 0.7 for both classifications. Moreover, using MPM as a benchmark, the comparison of the metrics obtained with the BTD < 0°C approach with those derived with the NN model indicates that the neural network performs a more accurate volcanic cloud detection for both considered test cases.

Besides the NN plume mask validation, we also compared the pixels which the NN model classified as affected by meteorological clouds (hereafter NNCM) with the SLSTR standard product for meteorological clouds.

Among the cloud masks available in the SLSTR L1RBT product, the *confidence_in_summary_cloud* mask (hereafter CSCM) is considered. The CSCM is a cloud mask which discriminates cloud pixels (*true*) and cloud-free pixels (*false*); it is an ultimate cloud mask product derived from several separated cloud tests (Polehampton et al., 2021). As the CSCM doesn't distinguish between meteorological liquid water clouds and meteorological ice clouds as the NN algorithm does, the comparison is realized by considering the whole NN meteorological cloud classes (i.e. the sum between *Cloud* and *Cloud_ice*).

Figure 8 displays the RGB composite, in which the Sentinel-3 sun glint mask is highlighted (right part of the scene), and the comparison between NN cloud mask and S3 cloud mask for S3A/SLSTR image collected on Raikoke for 22 June 2019 at 00:07 UTC (Figure 8(a) and Figure 8(b)) and for S3B/SLSTR image collected on Raikoke for 22 June 2019 at 23:01 UTC (Figure 8(c) and Figure 8(d)).

Also in this case, for the images displaying the comparison between the two types of cloud masks (Figure 8(b) and Figure 8(d)), green indicates the pixels classified as meteorological cloud for both procedures, while red and blue indicate the pixels classified as meteorological cloud only from the SLSTR standard product and NN, respectively. Pixels that are not coloured are associated to a cloud-free condition for both the NN and the S3 cloud mask. Looking at the comparison, a very good agreement between the NN meteorological cloud mask and the SLSTR standard cloud mask can be observed. The metrics in Table 5 show very good performances, reaching an F-measure around 0.9. Moreover, looking at the red pixels in the 23:01 UTC image especially, it can be noted that the SLSTR cloud mask also includes the volcanic cloud.

**Table 5:** NN meteorological cloud detection accuracy using classification metrics derived from the comparison between the NN cloud mask (NNCM) and the confidence in summary cloud mask (CSCM) for each SLSTR considered product which has been assumed as ground truth.

| Classified Product | Precision | Recall | F-measure | Accuracy |
|---|---|---|---|---|

| | | | | |
|---|---|---|---|---|
| S3A/SLSTR at 00:07 UTC | 0.891 | 0.936 | 0.913 | 0.842 |
| S3B/SLSTR at 23:01 UTC | 0.952 | 0.820 | 0.881 | 0.795 |

From the validation procedure we have carried out, a considerable point which has to be underlined is that, unlike adopting a time consuming and case-specific approach as MPM which also needs a manual operation by setting various thresholds for each case under examination, the NN model can be used to discriminate ash plume in satellite images with good accuracy in a fast and automatic way, which saves a significant amount of time by eliminating the need for manual intervention.

## 6 Conclusions

In this work the results of a new neural network based approach for volcanic cloud detection are described. The algorithm, developed to process Sentinel-3/SLSTR daytime images, exploits the use of MODIS daytime data as training. The procedure allows the full characterization of the SLSTR image by identifying, besides the volcanic cloud, surfaces under the cloud itself, meteorological clouds (and phases), land, and sea surfaces. As test cases, the S3A-S3B/SLSTR images collected over the Raikoke volcano area during the June 2019 eruption have been considered.

The proposed neural network based approach for volcanic ash detection and image classification shows an overall good accuracy for the ash class, which is the main target of the algorithm, and for the meteorological cloud class as well. A strong effectiveness of the NN classification is indeed also related to the cloudy pixel recognition, with the ability to distinguish two different types of meteorological clouds composed of water droplets and ice respectively. It has to be reminded that the wide distribution of meteorological clouds in the scenario under consideration makes the ash detection task particularly complex.

A point to be underlined is the valuable advantage of the procedure related to the creation of products (the eight classes) not all currently available as SLSTR standard products; this fact represents a considerable step forward for generation of novel types of S3/SLSTR products.

A post processing has been applied to NN outputs by exploiting the land/sea mask available in the SLSTR standard products, in order to mitigate the insurgence of NN land/sea failure.

The comparison between the NN plume mask and a reference plume mask (MPM) taken as *ground truth*, shows a good agreement between the two techniques. The significant result lies in the fact that the overall good performance of the NN output is achieved in an automatic way and with a brief processing time, compared to the plume mask specifically generated, which instead requires a longer time, is case-specific and needs the presence of an operator. The other considerable achievement of the NN developed procedure is that, once the NN model has been properly trained, it has been used to detect the ash plume for each SLSTR image related to the Raikoke eruption, while the creation of the MPM has to be made separately for each image.

The comparison between the NN cloud mask and the cloud mask derived from SLSTR standard products has also been carried out, resulting in a high percentage of agreement between the two products.

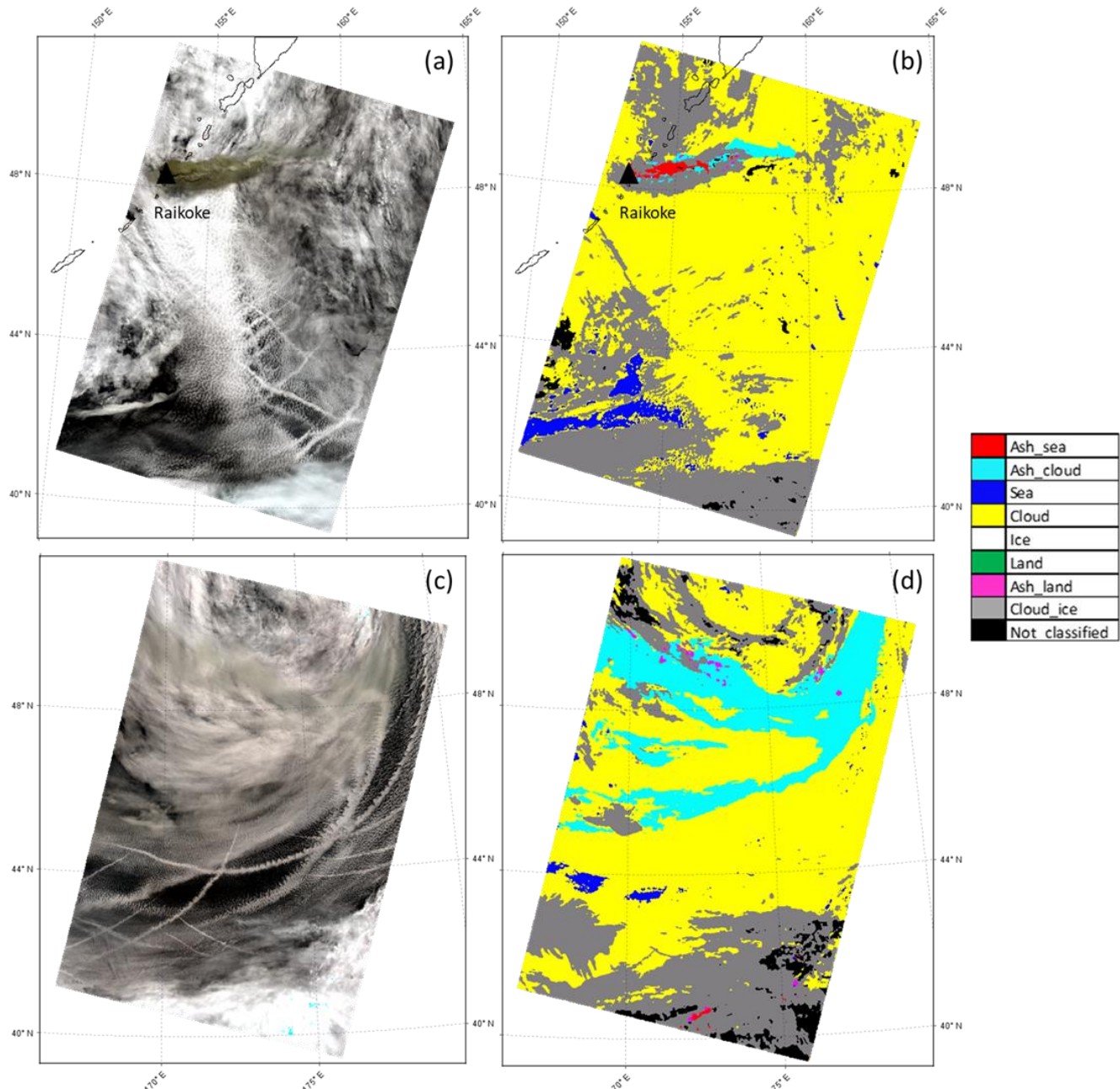

394

**Figure 9:** Sentinel-3A/SLSTR image collected on Raikoke for 22 Jun 2019 at 00:07 UTC, oblique view ((a) and (b)); Sentinel-3B/SLSTR image collected on Raikoke for 22 June 2019 at 23:01 UTC, oblique view ((c) and (d)). (a) and (c): RGB; (b) and (d): NN classification.

A promising outcome is related to the ability of the NN model to generalize over different data in terms of spatio-temporal and geographical characteristics, being the NN model trained with data collected over the Iceland region in 2010 and then applied to data acquired over the Kamchatka Peninsula in Russia in 2019. Something under consideration for future improvements is

to enhance the ability of the NN to generalize over various eruptive scenarios, by integrating different training dataset (in terms of regions, type of eruption, time interval, etc). In fact, the current methodology has been applied just to a few test cases and more validation is required in order to give the technique broader applicability. For example, the effects of varying moisture and atmospheric conditions has not been fully explored. On the other hand, the generation of an appropriate number of examples, which must be statistically representative of all the possible scenarios, to be included in the training dataset may represent a very difficult task. A possible approach could be the design of different neural networks, each associated with a specific scenario.

We also aim at further investigate some aspects in order to improve the classification accuracy, as the introduction of other output classes, such as volcanic ice cloud, and the integration of other variables in the model, such as the sensor view angle. Moreover, a fully comprehensive study about the sensitivity of the NN detection on the observation angle could be another possible future development of the study. Here we addressed briefly this point applying the trained network to SLSTR oblique view products, characterized by a zenith view angle of about 55° (Polehampton et al., 2021). shows the RGB composite and the NN classification for the SLSTR oblique view product collected on 22 June 2019 at 00:07 UTC ((a) and (b)) and 23:01 UTC ((c) and (d)) respectively. It is interesting, as a preliminary result, to show how, especially for the 23:01 UTC image where the opacity of the volcanic cloud is slighter, the main features of the classification map obtained using a NN model trained only on near nadir view acquired products and used for classifying oblique view data are mostly conserved. However, the complexity brought in by the difference in the slant optical depth, which may translate to a noticeable difference in top-of-atmosphere signal levels, needs to be investigated in a full dedicated study.

Finally, the possibility to use S3/SLSTR products to train a neural network able to detect volcanic clouds in Sentinel-3/SLSTR granules might improve the overall accuracy of the classification.

**Code availability**

The whole methodology is developed in MatLab environment. The source codes are available upon request to ilaria.petracca@uniroma2.it.

**Data availability**

Terra-Aqua/MODIS data are distributed from the Level-1 and Atmosphere Archive & Distribution System (LAADS) Distributed Active Archive Center (DAAC) and they are available at: https://ladsweb.modaps.eosdis.nasa.gov/search/.

Sentinel-3/SLSTR data are distributed from the Copernicus Open Access Hub and they are available at: https://scihub.copernicus.eu/dhus/#/home.

The dataset used for this study are freely available on the Zenodo platform (https://doi.org/10.5281/zenodo.7050771).

## Author contributions

IP and DDS developed algorithms, analyzed data and results and wrote the manuscript; MP developed algorithms and methodology, analyzed data and results and reviewed the manuscript; SC and LG analyzed data and results, provided reference data for validation task and wrote-reviewed the manuscript; FP supported the analysis of data and results, worked on the Himawari-8 analysis part of the manuscript, and reviewed the manuscript; LM and DS supported the analysis of data and results; FDF reviewed the manuscript, supervised the research and contributed to funding acquisition; GSal supported the analysis of data and results and worked on validation; GSch supports the research and contributed to funding acquisition. All authors have read and agreed to the published version of the manuscript.

## Competing interests

The authors declare that they have no conflict of interest.

## Disclaimer

Publisher's note: Copernicus Publications remains neutral with regard to jurisdictional claims in published maps and institutional affiliations.

## Special issue statement

This article is part of the special issue "Satellite observations, in situ measurements and model simulations of the 2019 Raikoke eruption (ACP/AMT/GMD inter-journal SI)". It is not associated with a conference.

## Acknowledgments

The results shown in this work were obtained in the framework of the VISTA (Volcanic monItoring using SenTinel sensors by an integrated Approach) project, which was funded by ESA within the "EO Science for Society framework" [https://eo4society.esa.int/projects/vista/].

## Financial support

Not applicable.

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
