# Peer review of "Volcanic cloud detection using Sentinel-3 satellite data by means of neural networks: the Raikoke 2019 eruption test case"

_Atmospheric Measurement Techniques, 2022_

## Author Comment (AC4)

**Response to Reviewer 3 comments**

**Interactive comments on "Volcanic cloud detection using Sentinel-3 satellite data by means of neural networks: the Raikoke 2019 eruption test case" by Petracca et al.**

We would like to thank the Reviewer for her/his constructive comments and suggestions, which have improved the manuscript.

Please find our replies to each comment below. Referee comments are reported in black. Our replies are given in red.
* * *
The Authors present a neural network technique to detect volcanic ash clouds by combining visible and thermal infrared channels of moderate resolution spectroradiometers. A neural network trained on MODIS imagery acquired during the Eyjafjallajökull eruption in 2010 is applied to two pairs (nadir and oblique view) of SLSTR images of the Raikoke eruption that occurred in 2019.

The neural network method is compared to the classic brightness temperature difference (BTD) method and the accuracy of the two methods is evaluated against manually classified pixels. The results show a reasonable performance of the NN method in detecting ash clouds in nadir view, whereas I have some perplexities about its performance in the oblique view, as it seems to me that the NN underperforms for a fairly thick ash plume which I would expect to be easily detectable.

All in all, I think the paper can be published with minor revisions, although I recommend careful proofreading by a native English speaker, as the quality of the written English does not look impeccable to me.

A native English speaker proofread the revised version of the paper.

**MAIN COMMENT**

I think that the extension of the NN approach to oblique view needs further investigation. What are the typical values of the viewing angles sampled in the oblique view and how do they compare to those of the nadir view? If the air mass sampled in the oblique view is much bigger than that sampled in nadir view, the difference in the slant optical depth may translate to a noticeable difference in top-of-atmosphere signal levels. Furthermore, if there is a large difference in the observed scattering angles you may be also sampling different ranges of (weather and ash) cloud phase functions, which also may lead to significant differences in the signal levels in VIS/NIR channels.

In this case, it looks far from obvious to me that the NN can still be applied reliably to oblique view situations that are probably not covered in the training set.

Therefore, I would recommend studying the sensitivity of the NN detection to the observation angle by generating synthetic top-of-atmosphere spectra of VIS/NIR radiance and thermal brightness temperature for a typical liquid water, ice and ash cloud. In my opinion, the results presented in the paper do not allow to draw reliable conclusions on the robustness of the NN method to off-nadir observations.

We thank the reviewer for all her/his interesting suggestions regarding the oblique view application.

Indeed, in case of the proposed work our intention was to preliminarily show an additional point with the idea to go in deep in future developments. For this reason, we moved the application of the NN model to the oblique view data in the conclusions section. As an anticipation we think it is interesting to show how the main features of the classification map (represented in Figure 7) obtained using a NN model trained only on near nadir view acquired products and used for classifying oblique view data are mostly conserved.

The complexity of the problem also involves the training dataset generation. In fact, below we report the histogram of the View Zenith Angles (VZA) used for MODIS Training (9 images) related to the pixels considered as ash. The VZAs greater than 40 degrees are undersampled with respect to the others and this could probably have an impact on the results of the off-nadir SLSTR view (SLSTR zenith angle in the oblique view is about 55°, as reported in figure below).

SLSTR zenith view angles for 00:07 UTC

SLSTR oblique view angles for 00:07 UTC

Finally, we modified the last sentence of the abstract to: "Finally, the results show that the NN developed for the SLSTR nadir view can produce reasonable image classification also for the SLSTR oblique view."

**DETAILED COMMENTS**

- P1, L15, spaceborne sensors acquired data -> satellite data

Done

- P1, L19, The classification of the clouds and of the other surfaces -> A classifications of clouds and other surfaces

**Done**

- P1, L22, foster the robustness of the approach, which allows overcoming  $\rightarrow$  allow to extend the approach to SLSTR, thereby overcoming

We changed to "...the robustness of the approach, which improves on the use of SLSTR products..."

- P2, L42, to detect the volcanic cloud -> to detect volcanic clouds

**Done**

- P2, L43, you can remove "problem" after detection

**Done**

- P2, L43, lies on -> relies on

**Done**

- P2, L44. There is no such thing as "water vapour clouds". I guess you mean "liquid water clouds"

Yes, we do mean that, but there are also "clouds of water vapour". However, we changed to "liquid water clouds".

- P2, L49, region -> regions

Done

- P2, L59, procedures described -> described procedures. Plus, is "among" really what you mean, or do you mean "in addition to"? Does "described" refer to Prata et al. (2001b) and Corradini et al. (2008,2009)?

Yes, thank you. We mean "in addition to" and "described" refers to the procedures mentioned in the introduction, including those in Prata et al. (2001b) and Corradini et al. (2008,2009). Now the text has been improved.

- P3, L70, statistical -> statistically

Done

- P3, L71. Is this real time capability really an advantage of the NN approach? Isn't the BTD method also in near real time, given that it involves taking a difference? Furthermore, in an emergency scenario is there really such a big advantage in correctly detecting a few more ash pixels than the BTD method?

The problem is not strictly related to the computation time of BTD which is actually very fast, but to the reliability and the time consumption associated to the choice of the threshold to be used, which is based on a subjective interpretation. Indeed, using simply BTD < 0 °C (as in standard procedure) not always gives good results. The choice of the BTD threshold needs more time (Radiative Transfer Model simulation) and the presence of an operator. We can say that the NN approach, keeping the operation fast, can be more reliable and objective compared with the BTD method in general.

- P3, L85-86, either "a vertically ascending cloud" or "vertically ascending clouds"

Done

- P4, L103, "water vapour" -> "liquid water"

Done

- P4, L107. At what angles does SLTR dual view observe?

SLSTR dual view observes at many angles. In general, the zenith angle in the oblique view is about  $55^{\circ}$  and the nadir view angles range as shown in the following plot (see also the plots in the reply to the main comment).

- P4, L109-110. I don't understand the use of "since" here. What do you mean when you say that the feasibility of the method was confirmed for high latitudes "since" your study area is at medium-high latitudes.

We now improved the understandability of the sentence removing "since our study area is located in medium-high latitudes" as it was unnecessary.

-P8, L159-160. How is each percentage in the confusion matrix computed? Furthermore, overall accuracy is not a particularly informative parameter. Given that the main focus is on ash, it may be useful to provide statistics on the task of ash detection (probability of detection, false alarm ratio, critical success index).

The accuracy percentages in the confusion matrix (Figure 4) are computed according to (Fawcett, 2006) and those values are related to the training phase with MODIS data. To evaluate the performances of the trained NN model for classifying the SLSTR products we inserted more informative indexes in Table 4 (and Table 5).

Ref:

Fawcett, T. (2006). An introduction to ROC analysis. Pattern Recognition Letters, 27(8), 861–874. https://doi.org/10.1016/j.patrec.2005.10.010

-P8, L166. What do you mean by "commission" and "omission" errors? I guess one is "false detection" and the other is "missed detection", but it is not clear which is which.

"Commission" and "omission" were changed with "False positives (false detection) and false negatives (missed detection)..".

-P9, Figure 5. If I look at panel a, it seems to me that on the edges of the plume there are quite a few pixels that the NN classifies as "cloud ice". Do you have an idea why this happens?

This could be due to pixel heterogeneity (some parts of the pixel are ash and some parts may be ice), or maybe there could be some cloud ice at the edges.

-P11, L186, emphasizes -> shows

Done

-P11, L198-199, "even if some pixels are misclassified as ash on land". For such a thick ash cloud I would indeed expect that there is hardly any information in the signal to distinguish ash over land from ash over sea or cloud. Does it really make sense to introduce such a fine distinction between ash classes? What do you gain from that?

We suppose that since we know where the land is, it makes logical sense to have two classes. In case of very thick ash cloud could be not relevant to know if it's "ash on land" or "ash on sea" or "ash on cloud". On the other hand, in other cases (semi-transparent or thin ash cloud), the differentiation in these 3 classes can be useful for many retrieval procedures.

-P11, L199, less false positives -> fewer false positives. On top of that, are they really false positives? Doesn't the BTD detect fewer ash pixels compared to the NN?

We agree on this point where actually we are comparing two methods, neither of which is considered as "truth". We modified the sentence to:

".. the NN algorithm is able to detect a wide volcanic cloud area and more ash, especially in the opaque regions, compared to the BTD approach".

-P11, L206, water vapour cloud -> liquid water cloud

Done

-P11, L213, aerial trails -> aircraft condensation trails

We changed to "aircraft contrails".

-P11, L214. What causes the BTD method to give false positives over contrails?

It shouldn't. We would expect contrails to have a positive BTD (that is the 11-12  $\mu$ m difference should be positive). It could happen if the ash is below the contrail and then the contrail might look like ash, or could be related to thermal contrast, and perhaps noise, pixel heterogeneity and viewing angle effects. In general, the broader question of false positives needs a deeper discussion.

Here a reference related to pitfalls with the BTD approach:

Prata, F. Bluth, G., Rose, W. I., Schneider, D. and A. Tupper (2001). Comments on "Failures in detecting volcanic ash from a satellite-based technique"., 78(3), 341–346. doi:10.1016/s0034-4257(01)00231-0

-P11, L218, "produces good results". I would say "reasonable". The ash cloud looks so thin here that I doubt you have a very good reference to compare your results against. How does the BTD approach perform for this image?

The role of this part has been completely transformed. Now is it reported in the Conclusions as a very preliminary result encouraging full dedicated studies addressing this topic.

However, we report below a figure showing the BTD map for the S3/SLSTR oblique view products for 00:07 UTC (left panel) and 23:01 UTC (right panel).

---

## Author Response (AR1)

The document is structured according to the following colour legend:

- comments from Referee;
- author's response
- author's changes implemented in the text.
* * *
**Response to Reviewer 1 comments**

**Interactive comments on "Volcanic cloud detection using Sentinel-3 satellite data by means of neural networks: the Raikoke 2019 eruption test case" by Petracca et al.**

We would like to thank the Reviewer for her/his constructive comments and suggestions, which have improved the manuscript.
Please find our replies to each comment below. Referee comments are reported in black. Our replies are given in red. The changes implemented in the text are marked in green.

**---**

This manuscript presents a neural network model in order to detect volcanic ash clouds using Sentinel-3 SLSTR (Sea and Land Surface Temperature Radiometer) daytime products. The neural network is trained with MODIS daytime imagery from the Eyjafjallajökull eruption in May 2010. Then it is applied to the Raikoke eruption in June 2019. The results show that the neural network model can accurately detect volcanic ash from Raikoke compared with RGB visual inspection and BTD (Brightness Temperature Difference) procedure. Moreover, the plumes identified by neural network model agree well with the plume manually identified for the specific SLSTR images.

The manuscript is very well structured and written. It addresses an important issue in detection of the volcanic ash clouds and presents a solution which is beneficial for remote sensing and modeling volcanic ash dispersion. The methods and assumptions are scientifically sound and the results are well elaborated. Thus, I recommend the manuscript for publication after adressing the following points:

1- The authors should use/cite the published data instead of relying on private communication (L92). Specifically, there are several papers in this special issue that present ash and SO2 mass (e.g. Muser et al 2020, ACP). I strongly suggest that the authors review the published papers related to Raikoke and use them in the introduction and discussions.

This part of the text has now been improved using additional references as reported below.

New references:

Bruckert, J., Hoshyaripour, G. A., Horváth, Á., Muser, L. O., Prata, F. J., Hoose, C., and Vogel, B.: Online treatment of eruption dynamics improves the volcanic ash and SO2 dispersion forecast: case of the 2019 Raikoke eruption, Atmos. Chem. Phys., 22, 3535–3552, https://doi.org/10.5194/acp-22-3535-2022, 2022.

Gorkavyi, N., Krotkov, N., Li, C., Lait, L., Colarco, P., Carn, S., DeLand, M., Newman, P., Schoeberl, M., Taha, G., Torres, O., Vasilkov, A., and Joiner, J.: Tracking aerosols and SO2 clouds from the Raikoke eruption: 3D view from satellite observations, Atmos. Meas. Tech., 14, 7545–7563, https://doi.org/10.5194/amt-14-7545-2021, 2021.

Muser, L. O., Hoshyaripour, G. A., Bruckert, J., Horváth, Á., Malinina, E., Wallis, S., Prata, F. J., Rozanov, A., von Savigny, C., Vogel, H., and Vogel, B.: Particle aging and aerosol–radiation interaction affect volcanic plume dispersion: evidence from the Raikoke 2019 eruption, Atmos. Chem. Phys., 20, 15015–15036, https://doi.org/10.5194/acp-20-15015-2020, 2020.

Prata, A. T., Grainger, R. G., Taylor, I. A., Povey, A. C., Proud, S. R., and Poulsen, C. A.: Uncertainty-bounded estimates of ash cloud properties using the ORAC algorithm: Application to the 2019 Raikoke eruption, Atmos. Meas. Tech. Discuss. [preprint], https://doi.org/10.5194/amt-2022-166, in review, 2022.

L105-106: "It is estimated from the AHI data that June 2019 Raikoke eruption produced approximately 0.4–1.8 Tg of ash (Bruckert et al., 2022; Muser et al., 2020; Prata et al., 2022) and 1–2 Tg of SO2 (Gorkkavyi et al., 2021; Bruckert et al., 2022)."

2- Raikoke and Eyjafjallajökull are both high-latitude volcanoes. How would the model perform on tropical eruptions like la Soufrière 2021? Is the model transferable to tropical conditions or different ash compositions? It will be interesting to see the application to la Soufrière.

Overall, the main purpose of the paper was to develop a methodology based on a neural network model able to classify SLSTR products for the Raikoke 2019 eruption, investigating the feasibility of training the model with MODIS data at comparable latitudes given the lack of SLSTR products for eruptions at such latitudes.

The complexity of the application suggests that the generalization of the methodology to all types of eruptions is not straightforward, and this was confirmed by some preliminary analysis (also including la Soufrière 2021). For example, the change of latitude has an impact on the characteristics of the atmosphere. At the same time different volcanoes emit different types of ash affecting the variability of the radiance values detected by the sensors. A possible solution to overcome the model transferability issue could be the training of different NN models for specific latitude belts which can be defined to cover the whole globe.

However, we inserted some comments in the discussion/conclusions dedicated to the uncertainties and limitations of the proposed model, as requested in point 4 also (see reply to point 4).

3- I would like to see the $R^2$ and RMSE of the neural networks during training, validation and test. The topology of the neural network model (large number of neurons in the hidden layer) and split of the training/validation/test might lead to overfitting. Besides, please add info about the training method.

In the text we added information about activation function, hardware and time needed for training the proposed model.

L250; L267-268

In Figure 4 we report the confusion matrix during for the validation set which is indicative of the model generalization capability of classification on data which have been not used for training and test the model. Training neural networks for classification problems the accuracy of the confusion matrix (90.9%) on the validation set can be considered as meaningful metric instead of the $R^2$, which is usually used mostly for regression problems.

Moreover, as the graph below shows, we avoided overfitting through the early stopping technique. The model used for classifying Raikoke SLSTR granules have been trained until epoch 53 where the minimum error on validation have been obtained (MSE = 0.0182).

[Figure]

4- There are no discussions on the uncertainty and the limitations of the presented model.

Comments on the uncertainty and the limitations of the presented model have been added in sections 5 and 6.

L332-344:
"Our results suggest that the NN technique is robust and has shown that it is possible to transfer the NN model from one single eruption event to others occurring at similar latitudes. However, the complexity of the application suggests that the generalization of the methodology to all types of eruptions is not straightforward. For example, the change of latitude has an impact on the characteristics of the atmosphere. At the same time different volcanoes emit different types of ash affecting the variability of the radiance values detected by the sensors. A possible solution to give to the proposed technique a broader applicability could be training different NN models for specific latitude belts which can be defined to cover the whole globe.
Overall, we can summarize the main uncertainties and the limitations of the presented model in the following points:

1. model transferability is significantly related to the spatial-temporal data availability for the generation of a training dataset which is statistically representative of all the possible scenarios;

2. lack of standard ground truth data for training and validation phases requires the BTD threshold selection by an operator which prevents the method from being fully objective."

L479-486:
"Something under consideration for future improvements is to enhance the ability of the NN to generalize over various eruptive scenarios, by integrating different training dataset (in terms of regions, type of eruption, time interval, etc). In fact, the current methodology has been applied just to a few test cases and more validation is required in order to give the technique broader applicability. For example, the effects of varying moisture and atmospheric conditions has not been fully explored. On the other hand, the generation of an appropriate number of examples, which must be statistically representative of all the possible scenarios, to be included in the training dataset may represent a very difficult task. A possible approach could be the design of different neural networks, each associated with a specific scenario."

**Specific comments:**

L32-34: this part is not precise. Ash is a part of tephra with D<2 mm. Then we have fine and very fine ash. Please revise.

This part of the text has now been improved as reported below.

L34-37:

"From the start of an eruptive event, volcanic emissions are composed of a broad distribution of ash particles, ranging from very fine ash (particle diameters, $d < 30\,\mu m$) increasing in size to tephra (airborne pyroclastic material) with diameters from 2 mm up to 64 mm. Larger fragments are also generated which fall out quickly; these and ash with $d > 30\,\mu m$ are not considered in this paper."

L49: you mean $\Delta T_{11\mu m\text{-}12\mu m}$?

Yes, now the style has been set properly.

L70-72: NNs are good for what they are trained for. Their transferability to other eruption at different altitudes and with different ash composition (optics) might be challenging. Please comment on this.

Yes, the model transferability might be challenging in case of different conditions. For example, the change of latitude has an impact on the characteristics of the atmosphere. At the same time different volcanoes emit different types of ash affecting the variability of the radiance values detected by the sensors. Therefore, the generation of an appropriate number of examples, which must be statistically representative of all the possible scenarios, to be included in the training dataset may represent a very difficult task. However, a possible approach could be the design of different neural networks, each associated with a specific scenario.

L159: What is the measure of accuracy? $R^2$?

The accuracy of the trained model on the MODIS validation dataset was 90.9% as reported in the confusion matrix in Figure 4. Using the proposed vicarious validation to evaluate the performance of the model on SLSTR data some metrics have been added to Table 4 and 5 (see also our reply to the final comment).

L375-378; L434-436

Regarding the $R^2$, please see the reply to the comment n.3 (page 3).

L205: for consistency, use "meteorological clouds" in the whole manuscript.

Now we use always "meteorological clouds" in the whole text.

L226: this argument is too strong. See my previous comments.

According to the previous comments and replies, this part of the text has now been improved.

L332-344

Tables 4 and 5: It is very difficult to make any quantitative conclusion from these tables. Use other quantitative measures like SAL.

We derived the following metrics to improve quantitative conclusions (added to Tables 4 and 5):

- Precision;
- Recall;
- F-measure;
- Accuracy.

Ref:

Fawcett, T. (2006). An introduction to ROC analysis. Pattern Recognition Letters, 27(8), 861–874. https://doi.org/10.1016/j.patrec.2005.10.010

L375-378; L434-436

**Response to Reviewer 2 comments**

**Interactive comments on "Volcanic cloud detection using Sentinel-3 satellite data by means of neural networks: the Raikoke 2019 eruption test case" by Petracca et al.**

We would like to thank the Reviewer for her/his constructive comments and suggestions, which have improved the manuscript.
Please find our replies to each comment below. Referee comments are reported in black. Our replies are given in red. The changes implemented in the text are marked in green.

**---**

The study "Volcanic cloud detection using Sentinel-3 satellite data by means of neural networks: the Raikoke 2019 eruption test case" by Petracca et al. introduces a scene classification algorithm for the Sentinel-3 Sea and Land Surface Radiometer data based on neural networks. The classification is applied in a case study of the eruption of the Raikoke volcano in 2019. While the focus is on detecting volcanic ash plumes the classification mask also provides information on the surface, underlying surface under volcanic ash, and clouds. Although the paper is well structured and written I miss substantial information on the neural network. No information on how it was coded nor the source were provided. Moreover the results presented in this study lack a comparison with already published findings on the Raikoke eruption and measurements by other instruments. Hence I'd recommend a major revision before publication.

**General comments:**

In the introduction solely volcanic ash measurements in the mid-infrared are discussed. However the SLSTR mainly has channels in the VIS to near infrared spectral range. I suggest to also introduce VIS/near-IR volcanic ash measurements.

The volcanic ash measurements discussed in the introduction specifically concern the Thermal Infrared Region (TIR) ranging from 7 to 14 µm, not the mid-infrared region. In the TIR region indeed we find the most important information for volcanic ash measurements, while the VIS/NIR channels do not provide added information.

L53: "(7-14 µm)"

Besides, the SLSTR instrument has the ATSR sensor as heritage and it was designed around the IR channels. The VIS/NIR channels were added to assist in detecting clouds for the main purpose of using the IR channels to derive SST, and the whole innovation of the dual view was to aid the derivation of SST from the IR channels.

Throughout the manuscript ``weather clouds'' are mentioned. Please specify what you mean. Ice clouds, liquid clouds, mixed phase clouds, or all?

"Weather clouds" stand for all types of meteorological clouds.

Throughout the manuscript, in the revised version we replaced "weather clouds" with "meteorological clouds".

The description of the case study on the Raikoke eruption lacks references. Please have a look at the publications in this special issue to verify your reconstruction of the plume (in Fig. 1) and to substantiate your estimates of SO2 and ash.

More details on the Raikoke eruption have been inserted and new references have been added. Please find the new references below:
- Bruckert, J., Hoshyaripour, G. A., Horváth, Á., Muser, L. O., Prata, F. J., Hoose, C., and Vogel, B.: Online treatment of eruption dynamics improves the volcanic ash and SO2 dispersion forecast: case of the 2019 Raikoke eruption, Atmos. Chem. Phys., 22, 3535–3552, https://doi.org/10.5194/acp-22-3535-2022, 2022.
- Gorkavyi, N., Krotkov, N., Li, C., Lait, L., Colarco, P., Carn, S., DeLand, M., Newman, P., Schoeberl, M., Taha, G., Torres, O., Vasilkov, A., and Joiner, J.: Tracking aerosols and SO2 clouds from the Raikoke eruption: 3D view from satellite observations, Atmos. Meas. Tech., 14, 7545–7563, https://doi.org/10.5194/amt-14-7545-2021, 2021.
- Muser, L. O., Hoshyaripour, G. A., Bruckert, J., Horváth, Á., Malinina, E., Wallis, S., Prata, F. J., Rozanov, A., von Savigny, C., Vogel, H., and Vogel, B.: Particle aging and aerosol–radiation interaction affect volcanic plume dispersion: evidence from the Raikoke 2019 eruption, Atmos. Chem. Phys., 20, 15015–15036, https://doi.org/10.5194/acp-20-15015-2020, 2020.
- Prata, A. T., Grainger, R. G., Taylor, I. A., Povey, A. C., Proud, S. R., and Poulsen, C. A.: Uncertainty-bounded estimates of ash cloud properties using the ORAC algorithm: Application to the 2019 Raikoke eruption, Atmos. Meas. Tech. Discuss. [preprint], https://doi.org/10.5194/amt-2022-166, in review, 2022.

L105-106: New references have been added to substantiate the produced amount of ash of 0.4-1.8 Tg (Bruckert et al., 2022; Muser et al., 2020; A. T. Prata et al., 2022), and the produced amount of $SO_2$ of 1-2 Tg (Bruckert et al., 2022; Gorkavyi et al., 2021).

The methodology section I found somewhat confusing. Maybe separate the instrument description from the method description. The description of both instruments, MODIS and SLSTR, lack some information. What is their spectral range? What is their equatorial crossing time? Since when are they operating? What is the oblique view of SLSTR, which is mentioned

later? Which data products were used? First I had the impression that the classification categories (Ash over sea, ash over clouds, sea surface, ...) are MODIS products. Only later I realized that you made up these categories manually from MODIS Eyjafjallajökull observations. Please improve the description.

A section (number 3) regarding instruments specifications has been inserted.

L122-163: New section number 3 "Instrument" with paragraph 3.1 "MODIS instrument" and 3.2 "SLSTR instrument" has been added in the corresponding lines.

The description of the classification categories has been improved and the lack of some of the species (i.e. classification classes) in MODIS standard products has been remarked in the text.

L191-198, L206, L232-234, L236: Improved description of the creation of the training set.

Concerning the neural network, how did you build the network? Did you use Python and some packages? Did you use anything else? Please provide more information.

As added in the Code Availability section, the procedure has been developed in MatLab environment. In particular, the MatLab Deep Learning Toolbox has been used to implement the NN. The code of the procedure ran with a CPU i7-9850H (6 core, processor base frequency at 2.60 GHz) and it takes less than 30 minutes to train the adopted model and few seconds to apply it.

All these information are now included in the text: L166-168, L267-268, L503-505.

Also you mention the time benefit of using NNs. How much time did it take to train the NN? How long does it take to analyse a scene with the NN compared to the BTD method? When mentioning the speed advantage, please provide numbers/measurements.

The problem is not strictly related to the computation time of BTD which is actually very fast, but to the reliability and the time consumption associated to the choice of the threshold to be used, which is based on a subjective interpretation. Indeed, using simply BTD < 0 °C (as in standard procedure) not always gives good results. The choice of the BTD threshold needs more time (Radiative Transfer Model simulation) and the presence of an operator. We can say that the NN approach, keeping the operation fast, can be more reliable and objective compared with the BTD method in general. NN is indeed able to make the detection of ashy pixels in automatic way, once properly trained (is the training that needs much time, but once done it the application is fast). The time needed to make the classification of ash and other classes of a SLSTR image with our model is of the order of few minutes.

When comparing the results from the BTD-method with the results of the NN-approach, please comment on the sensitivity of both methods (BTD and NN), as well as the manual detection in

the VIS, on the ash AOD. Why should the BTD-approach lead to false positives in the case of the Raikoke?

We added quantitative conclusions in Table 4 analysing NN and BTD < 0 °C compared to the Manual Plume Mask.

L375-378: Table 4 has been modified.

Probably the main reason for false detections is that there could be low thermal contrast. Detection of ash over cold surfaces can be an issue (ash cloud and underlying surface may have similar temperatures). Another potential issue for geo sensors only is that at high viewing zenith angles there is increased sensitivity up to a critical angle, after which there can be positive differences for ash. This can lead to both false positives and false negatives. It gets very complicated because the pixel size also increases which makes heterogeneity also an issue.
The manual detection is not be made with VIS, but with TIR channels and brightness temperatures, see next comments for detailed discussion.

L410-412: Comment on comparison between BTD < 0 °C and Manual Plume Mask, and NN and Manual Plume Mask.

I clearly disagree that Section 4.1 is a validation of the method. The reference is tuned towards an ash plume discernible in RGB satellite images. The detection sensitivity towards ash/aerosol AOD in nadir geometry and VIS spectral range is different to other wavelengths and satellite measurement geometries. Since the NN method relies on multiple wavelengths ranging from VIS to mid-IR, the results should be compared to VIS to mid-IR standard ash/aerosol detection products. Why don't you compare with measurements of other instruments, e.g. TROPOMI, AIRS, IASI, OMI, GOME-2, CALIPSO?

Although we acknowledge that our comparison is not perfect and pure, as far as we know there are no ash standard product and the manual plume mask we realized is the only way to obtain a benchmark for a quantitative pixel by pixel comparison. However, now we changed the name of that section to "Vicarious Validation".

L346: "Validation" has been changed in "Vicarious Validation".
L349 clarifies the lack of ash standard products.

Moreover, we think we could consider the Sentinel-5P/TROPOMI $SO_2$ product only for qualitative comparison (see figure and comments below), while a full reliability of an Ash Index or an Aerosol Index product may be debatable. As an example, we report below the Aerosol Index from TROPOMI, but the interpretation of that data appear more complex than the $SO_2$ layer in this case. There are many issues validating classification results against those obtained with other instruments (Corradini, S., Guerrieri, L., Brenot, H., Clarisse, L., Merucci, L., Pardini, F., ... & Theys, N. (2021). Tropospheric Volcanic SO2 Mass and Flux Retrievals from Satellite. The Etna

December 2018 Eruption. Remote Sensing, 13(11), 2225) for example the different acquisition time, the different pixel size, etc.

Moreover, it has to be clarified that the manual plume mask we realized and we took as reference is not tuned towards an ash plume discernible in RGB satellite images but it is obtained from TIR channels (BTD thresholds and brightness temperatures), (see l350-359 in the text for a detailed explanation of how the manual plume masks have been created for both the two SLSTR data to which the NN has been applied).

L231-233 has been changed from "The identification of the ashy pixel is pursued by creating a mask according to specific BTD thresholds (from 0.0 to -0.4 °C) and a manual correction performed through visual inspection of each MODIS image" to "The identification of the ashy pixel is pursued by creating a mask according to specific BTD thresholds (from 0.0 to -0.4 °C) for each MODIS image For this purpose, the MOD/MYD021KM product has been used to derive the brightness temperatures required to compute the BTD."

Here a qualitative comparison between S5P/TROPOMI SO$_2$ (upper left panel) and Aerosol Index 354_388 (upper right panel) products collected the 23 June 2019 at 02:03 UTC and NN plume mask for the S3/SLSTR data collected the 22 June 2019 at 23:01 UTC is shown (lower panel). The S5P/TROPOMI products have been georeferenced in the SLSTR grid (23:01 UTC image).

[Figure]

[Figure]

As we can observe, the NN plume mask derived from SLSTR image is reasonably similar to the SO$_2$ plume derived from TROPOMI. However, the output of our classification is not the SO$_2$ plume but the ash plume, even if they are connected to each other.

Moreover, the application of this method to only 2 scenes of a single volcanic eruption, measured on the same day is rather inconclusive. Please consider applying the NN method to other volcanic eruptions (as e.g. Gray and Bennartz, 2015, tested their NN approach to 7 volcanic eruptions). Also, how would you method deal with desert dust, which is a challenge to the BTD approach?

Overall, the main purpose of the paper was to develop a neural network model able to classify SLSTR products for the Raikoke 2019 eruption, investigating the feasibility of training the model with MODIS data at comparable latitudes given the lack of SLSTR products for eruptions at such latitudes. Thus, our work does not present a general and global algorithm for ash classification, but it can be considered a good starting point to develop a technique with broader applicability, for which a deeper investigation is needed. We considered this improvement in future steps, in particular we planned to build different NN models for different latitude belts which can be defined to cover the whole globe. We also have inserted some comments dedicated to the uncertainties and limitations of the proposed model in the section "Results and Discussion" and "Conclusions".

L332-344: Uncertainties and limitations of the proposed model.
L481-490: Uncertainties and limitations of the proposed model and future developments.

In order to introduce the desert dust class (we have already considered it as a future step) we need to create a dataset comprising pixels affected by desert dust, but in the scenes we considered the desert dust is absent.

**Specific comments:**

l33-34: Please specify coarse and fine in µm.

We revised and changed in the text.

L34-37: "In general, from the start of the eruption, volcanic emissions are composed of a broad distribution of ash particles, ranging from very fine ash (particle diameters, $d < 30\,\mu m$) increasing in size to tephra (airborne pyroclastic material) with diameters from 2 mm up to 64 mm. Larger fragments are also generated which fall out quickly; these and ash with $d > 30\,\mu m$ are not considered in this paper. […]"

l34-35: Volcanic plumes also have a liquid part, as formation of sulphate aerosol starts immediately e.g. see Glasow et al. (2009).

The presence of the liquid part has been inserted in the text.

L41

l60: When mentioning other volcanic ash detection algorithms, please also consider Gangale et al. (2010) and Clarisse et al. (2013).

We added the following reference which talks about volcanic ash retrieval methods:
Clarisse, L., & Prata, F. (2016). Chapter 11—Infrared Sounding of Volcanic Ash. In S. Mackie, K. Cashman, H. Ricketts, A. Rust, & M. Watson (Eds.), Volcanic Ash (pp. 189–215). Elsevier. https://doi.org/10.1016/B978-0-08-100405-0.00017-3

L70

l68: I wonder why you are referring to two studies using NNs for ozone retrievals, although sufficient examples for aerosol and clouds are already mentioned.

We referred to the general use of NNs in atmospheric science for parameters estimation, however those references have been removed according to your suggestion and we added Gray and Bennartz (2015).

L80-81

l87: What does near the vent mean? Please specify the radius around the volcano from which the BT of the plume was derived. Also what does ``some distance upwind'' mean? Was it always the same distance? Which criteria did you apply?

The coordinates of the box near the vent are:

lon1=153.25

lon2=153.35

lat1=48.32

lat2=48.42

and the coordinates upwind from the vent are:

lon1=153.10

lon2=153.20

lat1=48.32

lat2=48.42

The coordinates of the vent are: lon = 153.24167, lat = 48.29167

Here's an image showing the locations:

[Figure]

The location information have been included in the caption of Figure 1.

l92-94: Please remove speculations about the water vapour.

The paragraph has now been improved and new references about the presence of water vapour in eruptions have been added (listed below), in particular McKee et al., 2021 refers to lightning in the Raikoke eruption and notes the presence of water to enhance lightning strikes.

Rose, W. I., D. J. Delene, D. J. Schneider, G. J. S. Bluth, A. J. Krueger, I. Sprod, C. McKee, H. L. Davies and G. G. J. Ernst, 1995, Ice in the 1994 Rabaul eruption cloud: implications for volcano hazard and atmospheric effects, Nature, 375: 477- 479.

McKee, K., Smith, C. M., Reath, K., Snee, E., Maher, S., Matoza, R. S., … Perttu, A. (2021). Evaluating the state-of-the-art in remote volcanic eruption characterization Part I: Raikoke volcano, Kuril Islands. Journal of Volcanology and Geothermal Research, 419, 107354. doi:10.1016/j.jvolgeores.2021. (This reference refers to lightning in the Raikoke erption and notes the presence of water to enhance lightning strikes).

Murcray, D. G., F. J. Murcray, D. B. Barker, and H. J. Mastenbrook (1981), Changes in stratospheric water vapor associated with the Mount St. Helens eruption, Science, 211, 823–824.

Glaze, L. S., S. M. Baloga, and L. Wilson (1997), Transport of atmospheric water vapor by volcanic eruption columns, J. Geophys. Res., 102, 6099–6108, doi:10.1029/96JD03125

Sioris, C. E., A. Malo, C. A. McLinden, and R. D'Amours (2016), Direct injection of water vapor into the stratosphere by volcanic eruptions, Geophys. Res. Lett., 43, 7694–7700, doi:10.1002/ 2016GL069918.

Xu, J.; Li, D.; Bai, Z.; Tao, M.; Bian, J. Large Amounts of Water Vapor Were Injected into the Stratosphere by the Hunga Tonga– Hunga Ha'apai Volcano Eruption. Atmosphere 2022, 13, 912. https:// doi.org/10.3390/atmos13060912

Millán, L., Santee, M. L., Lambert, A., Livesey, N. J., Werner, F., Schwartz, M. J., et al. (2022). The Hunga Tonga-Hunga Ha'apai Hydration of the Stratosphere. Geophysical Research Letters, 49, e2022GL099381. https://doi. org/10.1029/2022GL099381

L108-112

l100: Please explain what is a ``multilayer perceptron neural network"?

A brief introduction to the MLP NN has been inserted in the methodology section.

L171-179: "The MLP NN model (Atkinson & Tatnall, 1997; Gardner & Dorling, 1998) consists in a multi-layer architecture with three or more types of layers. The first type of layer is the input layer, where the nodes represents the elements of a feature vector. The second type of layer is the hidden layer, and consists of only processing units. The third type of layer is the output layer and it represents the output data, which are the classes to be distinguished and are set to one (that of the chosen class) or zero (all other nodes) in image classification problems. All nodes (i.e. neurons) are interconnected and a weight is associated to each connection. Each node in each layer passes the signal to the nodes in the next layer in a feed-forward way, and in this passage the signal is modified by the weight. The receiving node sums the signals from all the nodes in the previous layer and elaborates them through an activation function before passing them to the next layer."

l108: What is the difference between Sentinel-3A and 3B?

Sentinel-3A and Sentinel-3B are two platform carrying the same instrument SLSTR, Sentinel-3B's orbit is identical to Sentinel-3A's orbit but flies +/-140° out of phase with Sentinel-3A. This information has been included in Section 3 regarding the details of the instruments.

L149-150

l109: Which procedure is meant here? I don't understand why this is mentioned after the instrument description.

This part has been removed given that it was already discussed in the Introduction.

L189-190: Deleted lines
L89-91: Discussion in the Introduction: "The use of MODIS as a proxy for SLSTR was already successfully tested in a previous work investigating the complex challenge of distinguishing ice and meteorological clouds (also containing ice) using neural networks on SLSTR data (Picchiani et al., 2018)".

Table 1: Please provide consistently the bandwidth for both instruments. Did you use all channels in the NN?

We add other information in Table 1, including bandwidth.

L201: Updated Table 1.

Yes, we used all the channels mentioned in Table 1 for the training and the application of NN model, this detail has been also remarked in the text.

L184-185

Fig. 2: Do the text ``Neural Network" and the picture mean the same, or are this two different neural networks? Also there are two arrows from SLSTR to both? networks leading to one classification. Are two different networks used for the classification?

The figure has been modified as below. Only one neural network has been used.

[Figure]

L205: Updated Figure 2

l116: What does ``nine MODIS data'' mean? Is it 9 days of data? Is it 9 swathes? Is it 9 images? Please indicate the lat/long region around Eyjafjallajökull that was selected.

 "nine MODIS data" has been modified in "nine MODIS granules" in the text. We mean 9 MODIS images.
The coordinates of the region around the Eyjafjallajökull considered for the training dataset generation are reported below:
   lon1=-15.28°
   lon2=-23.91°
   lat1=63.25°
   lat2=64.07°

L207: "nine MODIS data" to "nine MODIS granules"

l117: What does pattern mean? Is pattern=pixel?

One training pattern (i.e.: training example, i.e.: "ground truth") corresponds to one pixel of a specific target class as identified in MODIS images through the semi-automatic procedure. This means that we have several patterns for each class, which corresponds to the pixels associated to that class according to the semi-automatic procedure aforementioned. In particular, not all the pixels of the considered MODIS image are included in the training dataset (i.e.: the ensemble of the training patterns), but only a part of them are randomly included.
An explanation has been now introduced in the text.

L208-212

l133-141: Where and how large are the uncertainties of your ground truth? Are you considering the visual classification of RGB-images as the reference?

As already discussed in previous comment the manual plume mask we realized and we took as reference does not come from a visual classification of RGB-images but it is obtained from TIR channels (BTD thresholds and brightness temperatures).
Regarding the uncertainties of the ground truth, for what concerns the land and sea masks the uncertainty is almost null or however they have the same uncertainty of the MODIS land/sea mask product (since they are taken from it, in particular from MOD/MYD03 Level-1A Geolocation Fields). Also for the cloud mask the uncertainty can be considered equal to the corresponding MODIS product (MOD/MYD06_L2 Cloud Product) which have been used to create it. For the three ash classes and the ice class is more difficult to say the associated uncertainty.
The figure below shows the procedures used to create the training patterns for some target outputs as Plume_mask, Cloud_Mask, Land/Sea_Mask and Ice_Mask. The example is referred to one of the MODIS granule listed in Table 2.

[Figure]

l153-154: Is the a posteriori filter only applied to the categories ``land'' and ``sea'' or also to ``ash over land'' and ``ash over sea''?

The a posteriori filter is applied only to "land" and "sea" categories according to the land/sea mask available in the SLSTR data as standard product.

Fig. 5: What are the red and cyan color in the RGB image? Was the ``Not classified'' class only applied to ``Sea'' and ``Land'', or also to ``Ash_sea'' and ``Ash_land''?

The red in the RGB view, Figure 5(a), indicate the land according to the colour composite adopted (RED-S3, GREEN-S2, BLUE-S1), the cyan pixels in the RGB view are NaN value.
"Not classified" class is the result of the a posteriori filter, thus it is applied to "sea" and "land" categories.

l181: Does ``... difference between ... channels S8 and S9...'' mean mean radiance (S8) - mean radiance (S9)?

As explained in the text, we mean the difference between the brightness temperatures of the two channels S8 and S9. The S3/SLSTR channels from S7 to S9 are already provided as Brightness Temperatures in the S3/SLSTR product.

Fig. 6: Fig. 6a shows many contrails, but in Fig. 6c only few of them are classified as ``Cloud_ice''. Can you comment on this? Why are so many classified as ``cloud'' that was introduced as liquid cloud and which rather represents low altitude clouds?

As the NN has not received specific training information on contrails, the output classification over these objects may be not consistent.

l189: What do you mean by "pixels identified as volcanic cloud but that are not below the volcanic cloud..."? Please clarify.

We mean "Pixels which are identified by the NN model as belonging to the volcanic cloud while they actually are not part of the volcanic cloud", it means that they are easily recognizable as false detections of the BTD, i.e. false alarms.

L291-292: Rephrased.

l198-199: Here you state, that some pixels were misclassified as ``ash_land'' instead of ``ask_sea''. But shouldn't it rather be ``ash_cloud''? Most of the area around Raikoke is marked as ``cloud'' or ``ice cloud''. It would be surprising if only the region below the volcanic ash plume is not covered by clouds.

In the text we didn't state that the pixels classified by the NN as ash on land should instead be classified as ash on sea, we only state that the pixels classified by the NN as ash on land are misclassified. We have now improved this aspect the text.

L302

l206: What do you mean by ``water vapour cloud''? In the RGB images only ice, liquid water, or mixed clouds are visible.

Yes, we mean liquid water cloud class.

L310: "water vapour cloud class" to "liquid water cloud class".

l208: Having VIS RGB images at midnight sounds strange. I assume you mean 0 UTC.

We change in the text. We refer to the SLSTR image collected at 00:07 UTC.

L312

201-214: Why do you think the BTD approach produces wrong positive results in the case of the Raikoke eruption (Fig. 6)? Please explain. I'd rather consider the BTD ash plume realistic, because it pretty much resembles the $SO_2$ plume shape measured by TROPOMI on 23 June (e.g. Leeuw et al., 2021, Cai et al., 2022). How do you know that there wasn't any ash above the contrails and these underlying clouds enhanced the ash signal of the otherwise ``thin'' ash layer, which remained invisible in regions without underlying cold clouds (=high altitude clouds)?

In the text we referred to BTD false detections in Figure 6 only in relation to aircraft contrails (which actually are not included in the plume of $SO_2$ from TROPOMI, see left panel of the image below) and not in relation to the general shape of the BTD plume mask (see right panel of the image below), which we find indeed very similar to the TROPOMI $SO_2$ plume. However it has to be highlighted that we are comparing two methods (NN and BTD), neither of which can be considered as "truth".
For what concerns the presence of ash above the contrails we think that the underlying clouds would reduce the ash signal. Clouds (especially ice clouds – contrails) will have a positive BTD which will reduce or eliminate the negative BTDs (Prata, A. J. (1989a), Infrared radiative transfer calculations for volcanic ash clouds. Geophysical Research Letters, 16(11), 1293–1296. https://doi.org/10.1029/GL016i011p01293). The broader question of false positives is probably related to thermal contrast, and perhaps noise, pixel heterogeneity and viewing angle effects, and it needs a deeper discussion.
Here a reference related to pitfalls with the BTD approach:
Prata, F. Bluth, G., Rose, W. I., Schneider, D. and A. Tupper (2001). Comments on "Failures in detecting volcanic ash from a satellite-based technique". , 78(3), 341–346.      doi:10.1016/s0034-4257(01)00231-0

l220: Do you mean higher opacity here?

Yes, thank you. However the sentence has been rephrased and moved to the Conclusions.

Fig. 7c,d: Why are mostly clear regions (43-33N, 170-175E) classified as ``Cloud''? Please comment.

In case of the proposed work our intention was to preliminarily show an additional point with the idea to go in deep in future developments. For this reason we moved the application of the NN model to the oblique view data in the Conclusions section. As an anticipation we think it is interesting to show how the main features of the classification map (represented in Figure 7) obtained using a NN model trained only on near nadir view acquired products and used for classifying oblique view data are mostly conserved.
The complexity of the problem also involves the training dataset generation and this can produce error such as the one pointed out by the reviewer. In fact, below we report the histogram of the View Zenith Angles (VZA) used for MODIS Training (9 images) related to the pixels considered as ash. The VZA's greater than 40 degrees are undersampled with respect to the others and this could probably have an impact on the results of the off-nadir SLSTR view (SLSTR zenith angle in the oblique view is about 55°).

[Figure]

L321-332: Deleted SLSTR oblique view discussion.
L475-476, L489-497: SLSTR oblique view discussion has been inserted in the Conclusions.

l224-226: What do you mean by ``different scenario''? In terms of season, latitude, and injection height, the training eruption is similar to the showcase of the Raikoke eruption.

This part of the text has now been improved.

L332-344: Updated discussion about uncertainties and limitations of the proposed model.

Fig. 9: What does the white colour indicate? Why does the CSCM detect clouds in apparently clear regions?

White pixels in Figure 9 (b,d) indicate the areas for which both NN and CSCM don't detect the presence of cloudy pixels, as now has been introduced in the caption of Figure 9.

L396: In the caption of Figure 9 the description of the white pixels has been added.

The accuracy of CSCM (Cloud Mask product of S3/SLSTR) in detecting cloudy pixels is related to the already known limitations of the Confidence in Summary Cloud mask of S3/SLSTR product.

l274: Again, what are ``meteo clouds'' and ``meteo ice clouds''? Liquid and ice clouds?

Yes, "meteo clouds" are liquid water clouds and "meteo ice clouds" are ice clouds. We clarified in the text.

L419: "meteo clouds" to "meteorological liquid water", and "meteo ice clouds" to "meteorological ice clouds".

**Technical comments:**

l26-27: remove ``it'' -> ...which is...
Rephrased
L27

l27: manually -> manual
Done
L28

l30: NN, please introduce abbreviations
Deleted lines
L31-32: Deleted lines

l33: by -> of
Rephrased
L34-37

l49: region -> regions (2x)
Done
L57

l66: in -> at
Rephrased
L77

l84: AHI, please introduce abbreviations
Done
L97-98

Fig1 caption: was -> were; does -> do
Done
L117-118

l198: ash-on-land -> ash-over-land
Done
L301-302

l212: respect -> with respect
Done
L317

**References:**

Roland von Glasow, Nicole Bobrowski, Christoph Kern: The effects of volcanic eruptions on atmospheric chemistry, Chemical Geology, Volume 263, Issues 1–4, 2009, Pages 131-142, https://doi.org/10.1016/j.chemgeo.2008.08.020

G. Gangale, A.J. Prata, L. Clarisse: The infrared spectral signature of volcanic ash determined from high-spectral resolution satellite measurements, Remote Sensing of Environment, Volume 114, Issue 2, 2010, Pages 414-425, https://doi.org/10.1016/j.rse.2009.09.007

Clarisse, L., Coheur, P.-F., Prata, F., Hadji-Lazaro, J., Hurtmans, D., and Clerbaux, C.: A unified approach to infrared aerosol remote sensing and type specification, Atmos. Chem. Phys., 13, 2195–2221, https://doi.org/10.5194/acp-13-2195-2013, 2013.

Gray, T. M. and Bennartz, R.: Automatic volcanic ash detection from MODIS observations using a back-propagation neural network, Atmos. Meas. Tech., 8, 5089–5097, https://doi.org/10.5194/amt-8-5089-2015, 2015.

de Leeuw, J., Schmidt, A., Witham, C. S., Theys, N., Taylor, I. A., Grainger, R. G., Pope, R. J., Haywood, J., Osborne, M., and Kristiansen, N. I.: The 2019 Raikoke volcanic eruption – Part 1: Dispersion model simulations and satellite retrievals of volcanic sulfur dioxide, Atmos. Chem. Phys., 21, 10851–10879, https://doi.org/10.5194/acp-21-10851-2021, 2021.

Cai, Z., Griessbach, S., and Hoffmann, L.: Improved estimation of volcanic SO2 injections from satellite retrievals and Lagrangian transport simulations: the 2019 Raikoke eruption, Atmos. Chem. Phys., 22, 6787–6809, https://doi.org/10.5194/acp-22-6787-2022, 2022.

**Response to Reviewer 3 comments**

**Interactive comments on "Volcanic cloud detection using Sentinel-3 satellite data by means of neural networks: the Raikoke 2019 eruption test case" by Petracca et al.**

We would like to thank the Reviewer for her/his constructive comments and suggestions, which have improved the manuscript.
Please find our replies to each comment below. Referee comments are reported in black. Our replies are given in red. The changes implemented in the text are marked in green.
* * *
The Authors present a neural network technique to detect volcanic ash clouds by combining visible and thermal infrared channels of moderate resolution spectroradiometers. A neural network trained on MODIS imagery acquired during the Eyjafjallajökull eruption in 2010 is applied to two pairs (nadir and oblique view) of SLSTR images of the Raikoke eruption that occurred in 2019.

The neural network method is compared to the classic brightness temperature difference (BTD) method and the accuracy of the two methods is evaluated against manually classified pixels. The results show a reasonable performance of the NN method in detecting ash clouds in nadir view, whereas I have some perplexities about its performance in the oblique view, as it seems to me that the NN underperforms for a fairly thick ash plume which I would expect to be easily detectable.

All in all, I think the paper can be published with minor revisions, although I recommend careful proofreading by a native English speaker, as the quality of the written English does not look impeccable to me.

A native English speaker proofread the revised version of the paper.

**MAIN COMMENT**

I think that the extension of the NN approach to oblique view needs further investigation.
What are the typical values of the viewing angles sampled in the oblique view and how do they compare to those of the nadir view? If the air mass sampled in the oblique view is much bigger than that sampled in nadir view, the difference in the slant optical depth may translate to a noticeable difference in top-of-atmosphere signal levels. Furthermore, if there is a large difference in the observed scattering angles you may be also sampling different ranges of (weather and ash)
cloud phase functions, which also may lead to significant differences in the signal levels in VIS/NIR channels.
In this case, it looks far from obvious to me that the NN can still be applied reliably to oblique

view situations that are probably not covered in the training set.

Therefore, I would recommend studying the sensitivity of the NN detection to the observation angle by generating synthetic top-of-atmosphere spectra of VIS/NIR radiance and thermal brightness temperature for a typical liquid water, ice and ash cloud. In my opinion, the results presented in the paper do not allow to draw reliable conclusions on the robustness of the NN method to off-nadir observations.

We thank the reviewer for all her/his interesting suggestions regarding the oblique view application.

Indeed, in case of the proposed work our intention was to preliminarily show an additional point with the idea to go in deep in future developments. For this reason, we moved the application of the NN model to the oblique view data in the conclusions section.

L474-476; L489-497

As an anticipation, we think it is interesting to show how the main features of the classification map (represented in Figure 7) obtained using a NN model trained only on near nadir view acquired products and used for classifying oblique view data are mostly conserved.

The complexity of the problem also involves the training dataset generation. In fact, below we report the histogram of the View Zenith Angles (VZA) used for MODIS Training (9 images) related to the pixels considered as ash. The VZAs greater than 40 degrees are undersampled with respect to the others and this could probably have an impact on the results of the off-nadir SLSTR view (SLSTR zenith angle in the oblique view is about 55°, as reported in figure below).

[Figure]

[Figure]

[Figure]

Finally, we removed the last sentence of the abstract regarding the oblique view application.

**DETAILED COMMENTS**

- P1, L15, spaceborne sensors acquired data -> satellite data

Done

L15

- P1, L19, The classification of the clouds and of the other surfaces -> A classifications of clouds and other surfaces

Done

L19

- P1, L22, foster the robustness of the approach, which allows overcoming -> allow to extend the approach to SLSTR, thereby overcoming

Done

L22

- P2, L42, to detect the volcanic cloud -> to detect volcanic clouds

Done

L49

- P2, L43, you can remove "problem" after detection

Done

L51

- P2, L43, lies on -> relies on

Done

L51

- P2, L44. There is no such thing as "water vapour clouds". I guess you mean "liquid water clouds"

Yes, we do mean that and we changed to "liquid water clouds".

L52

- P2, L49, region -> regions

Done

L57

- P2, L59, procedures described -> described procedures. Plus, is "among" really what you mean, or do you mean "in addition to"? Does "described" refer to Prata et al. (2001b) and Corradini et al. (2008,2009)?

Yes, thank you. We mean "in addition to" and "described" refers to the procedures mentioned in the introduction, including those in Prata et al. (2001b) and Corradini et al. (2008,2009). Now the text has been improved.

L68-70

- P3, L70, statistical -> statistically

Done

L83

- P3, L71. Is this real time capability really an advantage of the NN approach? Isn't the BTD method also in near real time, given that it involves taking a difference? Furthermore, in an emergency scenario is there really such a big advantage in correctly detecting a few more ash pixels than the BTD method?

The problem is not strictly related to the computation time of BTD which is actually very fast, but to the reliability and the time consumption associated to the choice of the threshold to be used, which is based on a subjective interpretation. Indeed, using simply BTD < 0 °C (as in standard procedure) not always gives good results. The choice of the BTD threshold needs more time (Radiative Transfer Model simulation) and the presence of an operator. We can say that the NN approach, keeping the operation fast, can be more reliable and objective compared with the BTD method in general.

- P3, L85-86, either "a vertically ascending cloud" or "vertically ascending clouds"

Done

L99

- P4, L103, "water vapour" -> "liquid water"

Done

L181

- P4, L107. At what angles does SLTR dual view observe?

SLSTR dual view observes at many angles. In general, the zenith angle in the oblique view is about 55° and the nadir view angles range as shown in the following plot (see also the plots in the reply to the main comment).

[Figure]

- P4, L109-110. I don't understand the use of "since" here. What do you mean when you say that the feasibility of the method was confirmed for high latitudes "since" your study area is at medium-high latitudes.

This part has been removed given that it was already discussed in the Introduction.

L189-190: Deleted lines
L89-91: Discussion in the Introduction: "The use of MODIS as a proxy for SLSTR was already successfully tested in a previous work investigating the complex challenge of distinguishing ice and meteorological clouds (also containing ice) using neural networks on SLSTR data (Picchiani et al., 2018)".

-P8, L159-160. How is each percentage in the confusion matrix computed? Furthermore, overall accuracy is not a particularly informative parameter. Given that the main focus is on ash, it may be useful to provide statistics on the task of ash detection (probability of detection, false alarm ratio, critical success index).

The accuracy percentages in the confusion matrix (Figure 4) are computed according to (Fawcett, 2006) and those values are related to the training phase with MODIS data. To evaluate the performances of the trained NN model for classifying the SLSTR products we inserted more informative indexes in Table 4 (and Table 5).

Ref:

Fawcett, T. (2006). An introduction to ROC analysis. Pattern Recognition Letters, 27(8), 861–874. https://doi.org/10.1016/j.patrec.2005.10.010

L375-378; L434-436

-P8, L166. What do you mean by "commission" and "omission" errors? I guess one is "false detection" and the other is "missed detection",  but it is not clear which is which.

"Commission" and "omission" were changed with "False positives (false detection) and false negatives (missed detection).".

 L265

-P9, Figure 5. If I look at panel a, it seems to me that on the edges of the plume there are quite a few pixels that the NN classifies as "cloud ice". Do you have an idea why this happens?

This could be due to pixel heterogeneity (some parts of the pixel are ash and some parts may be ice), or maybe there could be some cloud ice at the edges.

-P11, L186, emphasizes -> shows

Done

L288

-P11, L198-199, "even if some pixels are misclassified as ash on land". For such a thick ash cloud I would indeed expect that there is hardly any information in the signal to distinguish ash over land from ash over sea or cloud. Does it really make sense to introduce such a fine distinction  between ash classes? What do you gain from that?

We suppose that since we know where the land is, it makes logical sense to have two classes. In case of very thick ash cloud could be not relevant to know if it's "ash on land" or "ash on sea" or "ash on cloud". On the other hand, in other cases (semi-transparent or thin ash cloud), the differentiation in these 3 classes can be useful for many retrieval procedures.

-P11, L199, less false positives -> fewer false positives. On top of that, are they really false positives? Doesn't the BTD detect fewer ash pixels compared to the NN?

We agree on this point where actually we are comparing two methods, neither of which is considered as "truth".  We modified the sentence to:

".. the NN algorithm is able to detect a wide volcanic cloud area and more ash, especially in the opaque regions, compared to the BTD approach".

L302-304

-P11, L206, water vapour cloud -> liquid water cloud

Done

L310

-P11, L213, aerial trails -> aircraft condensation trails

Done

L318

-P11, L214. What causes the BTD method to give false positives over contrails?

It shouldn't. We would expect contrails to have a positive BTD (that is the 11-12 µm difference should be positive).  It could happen if the ash is below the contrail and then the contrail might look like ash, or could be related to thermal contrast, and perhaps noise, pixel heterogeneity and viewing angle effects. In general, the broader question of false positives needs a deeper discussion.

Here a reference related to pitfalls with the BTD approach:
Prata, F. Bluth, G., Rose, W. I., Schneider, D. and A. Tupper (2001). Comments on "Failures in detecting volcanic ash from a satellite-based technique"., 78(3), 341–346.      doi:10.1016/s0034-4257(01)00231-0

-P11, L218, "produces good results". I would say "reasonable". The ash cloud looks so thin here that I doubt you have a very good reference to compare your results against. How does the BTD approach perform for this image?

The role of this part has been completely transformed. Now is it reported in the Conclusions as a very preliminary result encouraging full dedicated studies addressing this topic.

L489-497

However, we report below a figure showing the BTD map for the S3/SLSTR oblique view products for 00:07 UTC (left panel) and 23:01 UTC (right panel).

[Figure]

-P12, Fig. 7(a). Here the NN seems to detect a much smaller portion of the plume compared to what happens in the nadir image. Interestingly, a large fraction of the ash cloud is again classified as cloud ice (see my previous comment about Fig. 5).
Do you have any explanation for this apparently systematic tendency to confuse ash with cloud

ice? Again, how does the BTD approach perform for this case?

As in the reply to the previous comment, the role of this part has been completely transformed. Now is it reported in the Conclusions as a very preliminary result encouraging full dedicated studies addressing this topic.

As in the reply to the comment P9 about Fig. 5, this could be due to pixel heterogeneity (some parts of the pixel are ash and some parts may be ice), or maybe there could be some cloud ice at the edges.

Please see the figure attached to the previous comment for the BTD approach for the image.

-P12, L220, "this is due to the opacity of the volcanic cloud". Why is the opacity of the volcanic cloud a problem for the NN detection? I would expect a more opaque cloud to provide a better contrast against weather clouds.
Even visually, the plume in image (a) looks easier to detect than the faint plume in image (c).

An opaquer volcanic cloud could be easier to detect in the RGB bands, but this does not mean that it is easier to be discriminated from other species, in particular meteorological clouds, in the other spectral channels, which are used as input to the neural network. While, usually, the information in these other bands is crucial to resolve ambiguities, in our opinion this is a rather anomalous case (SLSTR image with volcanic cloud particularly thick and wide distribution of meteorological clouds) where the information coming from the infrared may generate some confusion in the NN output. In fact, as in the figure of the reply to comment P11 (L218), the BTD approach has a similar issue. The NN training dataset relies also on BTD and this also has an impact on the resulting classification.

-P13, L249, matching -> agreement

Done

L401

- P14, Table 4. How would such a table compare to a similar one for BTD vs MPM?

We added in Table 4 the classification metrics derived from the comparison BTD<0°C vs MPM in addition to those obtained from the comparison NN vs MPM.

L375-378

- P16, L286, includes also -> also includes

Done

L432